# Restoring cellular magnesium balance through Cyclin M4 protects against acetaminophen-induced liver damage

Acetaminophen overdose is one of the leading causes of acute liver failure and liver transplantation in the Western world. Magnesium is essential in several cellular processess. The Cyclin M family is involved in magnesium transport across cell membranes. Herein, we identify that among all magnesium transporters, only Cyclin M4 expression is upregulated in the liver of patients with acetaminophen overdose, with disturbances in magnesium serum levels. In the liver, acetaminophen interferes with the mitochondrial magnesium reservoir via Cyclin M4, affecting ATP production and reactive oxygen species generation, further boosting endoplasmic reticulum stress. Importantly, Cyclin M4 mutant T495I, which impairs magnesium flux, shows no effect. Finally, an accumulation of Cyclin M4 in endoplasmic reticulum is shown under hepatoxicity. Based on our studies in mice, silencing hepatic *Cyclin M4* within the window of 6 to 24 h following acetaminophen overdose ingestion may represent a therapeutic target for acetaminophen overdose induced liver injury.

Magnesium, or $Mg^{2+}$ in its ionized form, is one of the most abundant divalent cations in the cell and acts as a cofactor in hundreds of enzymatic reactions within the cell[1]. In mammals, the total cellular $Mg^{2+}$ concentration is maintained in the mid-millimolar range[2]. The essential role that $Mg^{2+}$ plays in mediating biochemical reactions at a cellular level explains why alterations in $Mg^{2+}$ homeostasis in the body lead to various disorders, such as the development of inflammatory responses in diabetes and asthma, preeclampsia, atherosclerosis, or heart damage[3]. Particularly in the liver, $Mg^{2+}$ deficiency has been associated with the risk of developing NASH[4], cirrhosis, alcoholic-derived liver damage, and hepatocellular carcinoma[5]. Restoring the correct balance of hepatic $Mg^{2+}$ thus emerges as an attractive strategy for treating liver pathology. However, no effective approaches for adjusting intracellular $Mg^{2+}$ concentrations are currently available[6].

At present, the contribution of dysregulated $Mg^{2+}$ homeostasis to the development of drug-induced liver injury (DILI) remains unknown. Due to the unpredictable nature and poorly understood pathogenesis of DILI, it has become a significant health problem, estimated to affect 19 out of 100,000 citizens worldwide[7]. In this context, the abuse of acetaminophen (N-acetyl-p-aminophenol, paracetamol, APAP), the

most commonly used drug in the USA, is responsible for almost 500 deaths annually from DILI, 100,000 calls to the US Poison Center, 50,000 emergency room visits, and 10,000 hospitalizations[8]. APAP overdoses account for 46% of all acute liver failures (ALF) in the USA and 40–70% of the cases in Europe[9].

To date, the only approved pharmacological treatment for APAP overdose and early stages of idiosyncratic DILI is N-acetylcysteine (NAC)[10,11]. However, NAC treatment is only effective within 20 h of the overdose[12] and standard doses of NAC in patients with advanced liver injury may not be sufficient[13]. Therefore, new therapeutic approaches are needed in this field[14].

N-acetyl-p-benzoquinone imine (NAPQI) represents the most harmful molecule derived from APAP metabolism. Its overproduction depletes hepatic reduced glutathione and enhances oxidative stress, which can lead to liver necrosis[12]. In this context, perturbation of the cross-talk between mitochondria and endoplasmic reticulum (ER) plays a major role in the development of hepatocellular death. Both organelles share cellular events such as c-Jun N-terminal kinase (JNK) pathway activation, essential in ER, and mitochondrial oxidative stress. Alterations in JNK activity modify the permeability of mitochondrial

✉e-mail: dbuccella@nyu.edu; amartinez@cicbiogune.es; mlmartinez@cicbiogune.es

membrane associated with alterations in calcium ($Ca^{2+}$) metabolism, perturbations in mitochondrial membrane potential, and loss of the ability to synthesize ATP[15]. Mitochondria-associated ER membranes (MAMs) are common specific domains that play a critical role in the local transfer of $Ca^{2+}$ to maintain cellular functions[16]. APAP intoxication leads to calcium depletion and the accumulation of misfolded proteins that activate an adaptive cellular response in this organelle[17]. In this context, protein homeostasis is maintained through the activation of two cellular processes, the so-called unfolded protein response (UPR) and autophagy- and proteasome-dependent proteolysis[18]. UPR activation involves three distinct pathways: inositol requiring 1α (IRE1α), PKR-like ER kinase (PERK), and activating transcription factor-6α (ATF6α)[15]. Importantly, the initial steps are characterized by increased gene expression of ER chaperones such as calreticulin, which binds $Ca^{2+}$ in the ER, and the $Ca^{2+}$ ATPase 2 (SERCA2), which pumps $Ca^{2+}$ from cytosol into the ER[19]. Some authors suggest that ER stress is a late event after APAP overdose[20,21], whereas mitochondrial alterations, ATP depletion, JNK activation, oxidative stress, and miss-regulation of $Ca^{2+}$ homeostasis are earlier events[20,22,23].

Previous studies[24] showed that mitochondria represent major intracellular reservoirs of $Mg^{2+}$. Mitochondrial $Mg^{2+}$ release in whole cells occurs upon mitochondrial depolarization[24]. Perturbation of this mitochondrial $Mg^{2+}$ levels then interferes with cellular energy metabolism and ATP production, sensitizing the cell to stress response[12].

Considering the protective properties of $Mg^{2+}$ and the key role of mitochondria and ER in acute liver failure caused by APAP overdose, we evaluated the homeostasis of this cation under these conditions. The flux of $Mg^{2+}$ across cell membranes is controlled by the activity and interplay of several $Mg^{2+}$ transporters, such as mitochondrial RNA splicing 2 (MRS2)[25], cyclin M 1-4 (CNNM1-4)[26], transient receptor potential melastatin 6-7 (TRPM6-7)[27], membrane magnesium transporter (MMgT)[28] and magnesium transporter subtype 1 (MagT1)[28].

Here, we show that in primary hepatocytes and in preclinical rodent models challenged with APAP, only the cyclin M4 (CNNM4) appears to be upregulated among all of these $Mg^{2+}$ transporters. Importantly, patients who underwent APAP DILI also displayed high levels of hepatic CNNM4 expression and perturbation of $Mg^{2+}$ serum levels. Although it is well established that CNNM4 plays a role in $Mg^{2+}$ transport across cell membranes, its specific role in the liver and possible contribution to hepatic pathology remain largely unknown[29]. In the present work, we have investigated the contribution of CNNM4 to the pathology of AILI in in vitro and in vivo models. Suppression of Cnnm4 expression in the liver restores hepatic $Mg^{2+}$ content in mitochondria and ensures proper calcium influx into the ER, resulting in downregulated activation of the JNK pathway shared by both organelles. Conversely, overexpression of Cnnm4 increases susceptibility to liver injury, whereas the T495I mutant of CNNM4[30], unable to efflux $Mg^{2+}$, lacks this capability. APAP toxicity boosts the CNNM4 in the ER cellular localization. Thus, regulation of $Mg^{2+}$ homeostasis by attenuating Cnnm4 expression protects mitochondria and ER stress, avoids necrosis, and consequently preserves liver function and protects against ALF. These results point to a mechanism that has been underestimated in liver pathology, focusing on $Mg^{2+}$ transport and the consequences of its modulation during DILI.

## Results

### Disturbance of $Mg^{2+}$ homeostasis and hepatic overexpression of CNNM4 are observed in patients with DILI and in preclinical models of APAP-induced DILI

$Mg^{2+}$ homeostasis has been linked to liver function[5,31]. To investigate whether $Mg^{2+}$ homeostasis could be dysregulated in patients with acetaminophen DILI, we evaluated $Mg^{2+}$ serum levels in a cohort of these patients, characterized in Supplementary Table I. $Mg^{2+}$ levels were significantly increased in patients with DILI ($n = 24$) versus

controls ($n = 4$) (Fig. 1a), suggesting that the flux of this cation may be aberrant in this pathological condition.

Preclinical animal models of APAP overdose have been developed that recapitulate many important aspects of human pathology. Evaluation of $Mg^{2+}$ also shows increased serum levels in animals treated with a single dose of APAP 360 mg/kg for 48 h compared with healthy animals (Fig. 1b). These results support the original hypothesis that there is a link between DILI and disturbances in $Mg^{2+}$ homeostasis. Despite significant progress in our knowledge of $Mg^{2+}$ transport and homeostasis, more research is required to characterize the specific transporters implicated in $Mg^{2+}$ flux in different physiological conditions. Several proteins have been identified as regulators of $Mg^{2+}$ homeostasis in vertebrates, including magnesium transporter 1 (MagT1), mitochondrial RNA splicing 2 (MRS2), and transient potential receptor melastatin 6-7 (TRPM6-7), as well as the ancient cation divalent protein/cyclin M 1-4 (CNNM1-4) family. A full characterization of the liver profile of these transports was performed in mouse primary hepatocytes and human hepatocyte THLE-2 cells exposed to APAP for different time periods, as well as in a mouse model of APAP-induced liver injury. We found that Cnnm4 at the mRNA level was remarkably increased in APAP overdose in both in vitro and in vivo models (Fig. 1c–e). CNNM4 protein and mRNA expression levels were also significantly elevated in the livers of mice after APAP-induced liver injury (Fig. 1f–h and Supplementary Fig. 2A). To demonstrate the specificity of CNNM4 staining in liver sections, competitive analyses were performed with the pure protein. The results shown in Supplementary Fig. 2B confirm the findings.

Given that the CNMMs were previously reported to be involved in maintaining $Mg^{2+}$ homeostasis[26,32], we investigated whether CNNM4 could also be associated with susceptibility to hepatic toxicity in humans. We measured its expression in liver biopsies from control individuals and patients diagnosed with DILI as characterized in Supplementary Table II. Quantitative histological analysis demonstrated increased hepatic CNNM4 in patients with DILI relative to healthy controls (Fig. 1i).

Overall, we provide strong evidence that hepatic CNNM4 expression is increased in DILI, which is also correlated with serum $Mg^{2+}$ levels.

### Targeting Cnnm4 resolves APAP toxicity in mouse and human hepatocytes

Cell death is a main characteristic of DILI. APAP overdose triggers cell inflammation and cell death in the liver[33]. We observed that Cnnm4 silencing reduced the extent of APAP-induced cell death, as assessed by TUNEL assay, while the remaining Cnnms failed to provide protection against the hepatocyte injury (Fig. 2a, Supplementary Fig. 3A, B). Importantly, the silencing of Cnnm4 does not affect the other Cnnms (Supplementary Fig. 3C). Moreover, the protective role of CNNM4 silencing was also supported by trypan blue viability assay at 3 and 6 h of APAP overdose (Supplementary Fig. 3D). To support these initial findings a de novo in silico design and screen for highly active and specific Cnnm4 siRNA molecules was performed. The best candidates were further characterized in vitro and in vivo.

Silencing of Cnnm4 with two different siRNAs in human hepatocyte THLE-2 cells also reduced cell death assessed 3 h after APAP treatment, evaluated by TUNEL and annexin V apoptosis and necrosis assays (Fig. 2b), and 6 h after treatment by TUNEL (Supplementary Fig. 4A).

In addition, we employed GalNAc siRNA technology developed for gene silencing, in hepatocytes. Cnnm4 siRNAs conjugated to triantennary N-acetylgalactosamine (GalNAc) are readily targeted to hepatocytes in vitro or in vivo by binding to the asialoglycoprotein receptor (ASGPR), which promotes specific and functional siRNA delivery by receptor-mediated uptake[34].

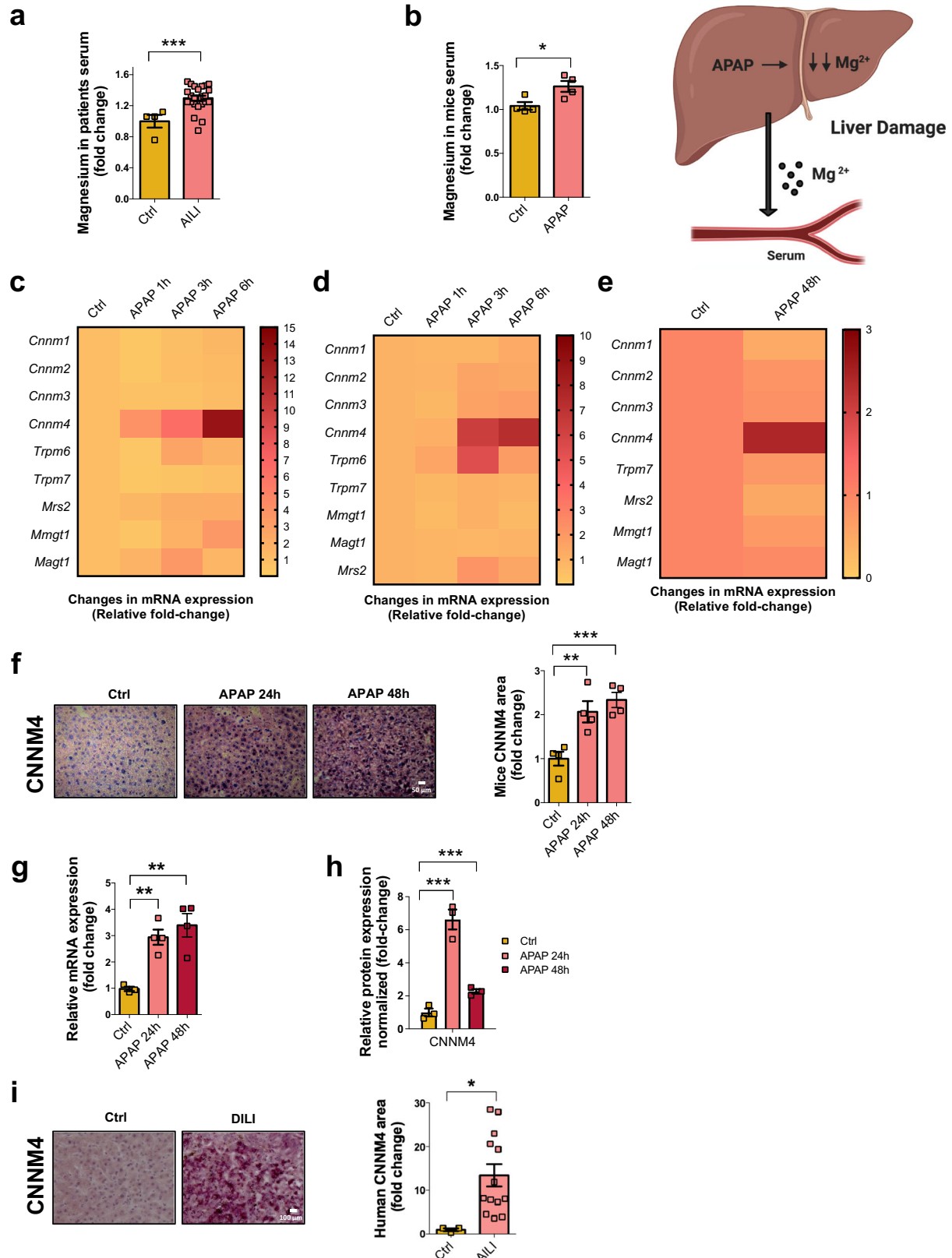

Of relevance, the two GalNAc-conjugated *Cnnm4* siRNAs employed completely attenuated the induction of *Cnnm4* expression at two different concentrations by APAP, which led to a significant reduction of cell death at 3 h, as evaluated by TUNEL and annexin V apoptosis and necrosis assays (Fig. 2c) and at 6 h after APAP overdose evaluated by trypan blue (Supplementary Fig. 4B) and TUNEL

(Supplementary Fig. 4C). Importantly, the complementary approach, inducing CNNM4 expression by transfection of Cnnm4 cDNA (pDEST-26 expression plasmid), boosted the sensitivity of primary hepatocytes to APAP overdose, leading to higher values of TUNEL and annexin V apoptosis and necrosis assays (Fig. 2d and Supplementary Fig. 4D).

**Fig. 1 | Cyclin M4 (CNNM4) is overexpressed in DILI patients and preclinical animal models caused by APAP. a** Magnesium serum levels determination in a cohort of AILI patients ($n = 24$) compared to a group of healthy donors ($n = 4$). Data are shown as mean ± SEM. ***$P < 0.001$ ($P = 0.001$) (Student's test, two-sided) **b** Magnesium serum levels was determined in a control group animals ($n = 4$). and those treated with a single dose of APAP 360 mg/kg for 48 h ($n = 4$). Data are shown as mean ± SEM. *$P < 0.05$ ($P = 0.033$) (Student's test, two-sided). Changes in mRNA levels of *Cnnm1, Cnnm2, Cnnm3, Cnnm4, Trpm6, Trpm7, Mrs2, Mmgt1,* and *Magt1* in (**c**) WT hepatocytes under APAP 10 mM compared to a control group (Ctrl) for 1 h, 3 h, and 6 h (**d**) THLE2 cell line (**e**) and the experimental groups of APAP 360 mg/kg ($n = 4$) mice compared to a healthy group ($n = 4$) at 48 h. Data are shown as mean. **f** Liver immunohistochemical staining and respective quantification of CNNM4 were determined in the experimental groups of APAP 360 mg/kg for 24 h ($n = 4$) and 48 h ($n = 4$) animal models compared to a healthy group ($n = 4$). Scale bar correspond to 50 μm. Data are shown as mean ± SEM. **$P < 0.01$, **$P < 0.001$ ($P = 0.01$ APAP 24 h vs Ctrl; $P = 0.001$ APAP 48 h vs Ctrl) (Student's test, two-sided) **g** mRNA levels *Cnnm4*. Data are shown as mean ± SEM. **$P < 0.01$ ($P = 0.0024$ APAP 24 h vs Ctrl; $P = 0.0059$ APAP 48 h vs Ctrl) (Student's test, two-sided) and **h** protein expression levels of CNNM4 in a animals treated with a single dose of APAP 360 mg/kg for 24 h ($n = 4$) and 48 h ($n = 4$) compared to a healthy group ($n = 4$), β-ACTIN was used as a loading control. Data are shown as mean ± SEM. ***$P < 0.001$ ($P = 0.00098$ APAP 24 h vs Ctrl; $P = 0.0009$ APAP 48 h vs Ctrl) (Student's test, two-sided) **i** Liver immunohistochemical staining and respective quantification of CNNM4 were determined in a cohort of AILI ($n = 13$) patients compared to a healthy group ($n = 3$). Scale bar corresponds to 100 μm. Values are represented as mean ± SEM. *$P < 0.05$ ($P = 0.038$) (Student's test, two-sided). Source data are provided as a Source Data file. The image was created with BioRender.com.

Considering the previous role described by CNNM4 as an extruder of Mg$^{2+}$ in non-hepatic cells and the beneficial effect of this cation in liver disease[35,36], primary hepatocytes with and without Cnnm4 overexpression were supplemented with Mg$^{2+}$ to mimic the potential therapeutic effect of this cation (Fig. 2d). Mg$^{2+}$ supplementation at different concentrations was not able to counteract the damage caused by APAP (Supplementary Fig. 4E) or Cnnm4 overexpression, measured by TUNEL and annexin V apoptosis and necrosis assays (Fig. 2d).

Previous data showed that amino acid substitution of CNNM4-T495I in the CBS domains of CNNM4 completely abrogates Mg$^{2+}$ efflux[30]. In order to understand its effect in primary hepatocytes under APAP treatment, the Cnnm4-T495I mutant was overexpressed. Importantly, Cnnm4-T495I overexpression did not induce any apoptotic response, based on the lack of inhibition in efflux activity (Fig. 2e and Supplementary Fig. 4F, G).

Finally, to evaluate APAP metabolism and bioactivation after *Cnnm4* knockdown, the activity of CYP2E1 was evaluated in primary hepatocytes after 3 h of treatment. APAP significantly reduced CYP2E1 activity, as described previously[37], and high levels of Cnnm4 showed the same tendency (Supplementary Fig. 4H). However, CYP2E1 activity was recovered in the absence of *Cnnm4*, while no type of modulation was observed with Cnnm4-T495I (Supplementary Fig. 4H).

Together, these data indicate that silencing of *Cnnm4* confers resistance to APAP-mediated necrosis, whereas overexpression of Cnnm4 confers sensitization to liver injury. Mg$^{2+}$ supplementation failed to counteract APAP-mediated toxicity in the presence of Cnnm4. The death mutant Cnnm4-T495I showed no effect on cell death.

## Targeting Cnnm4 restores Mg$^{2+}$ flux and reduces mitochondrial dysfunction in primary hepatocytes

Mitochondrial dysfunction triggered by APAP is one of the mechanisms that causes cell death in the liver[38]. First, we observed that the mitochondrial potential membrane in primary hepatocytes was restored in the absence of *Cnnm4* (Fig. 3a). Mitochondrial OXPHOS[39] monitored by seahorse analysis of hepatocytes under APAP exposure and *Cnnm4* silencing showed a higher oxygen consumption rate (OCR) in these conditions and reestablished the reservoir of ATP linked to respiration, avoiding the liver necrosis mediated by APAP overuse (Fig. 3b, c). Moreover, the absence of *Cnnm4* also resulted in a marked reduction in ROS production, assessed by MitoSOX, in APAP-treated primary hepatocytes (Fig. 3d). However, Cnnm4 overexpression induced mitochondrial ROS (Fig. 3e), while Cnnm4-T495I reduced ROS cellular content in comparison with APAP (Fig. 3e).

The function of CNNM4 in the liver under metabolic disturbances has been addressed[29]. Some authors have described CNNM4 in the intestinal epithelium as an extruder that exchanges intracellular Mg$^{2+}$ with extracellular Na$^+$, while others have pointed to its role as a Mg$^{2+}$ mediator of other transcellular transporters[35,40]. In order to further assess the relevance of the Mg$^{2+}$ flux in hepatocytes under *Cnnm4* silencing conditions, relative intracellular Mg$^{2+}$ levels were analyzed by mitochondrial-specific (Mg-S-TTP-AM)[29] and cytosolic-specific (Mg-S-AM)[29,41] labeling in primary hepatocytes after APAP administration (Fig. 3f). As expected, Mg$^{2+}$ content was decreased under hepatotoxic exposure, mainly in the mitochondria (Fig. 3f). On the other hand, *Cnnm4* silencing promoted restoration of normal Mg$^{2+}$ levels in these organelles and in the cytoplasm, indicating that in the liver CNNM4 functions as a Mg$^{2+}$ extruder (Fig. 3f). Mg$^{2+}$ content was also measured in the presence of 5 and 20 mM of this cation under APAP treatment. Mg$^{2+}$ mitochondrial content was not reestablished under these conditions (Fig. 3g). Finally, Cnnm4 overexpressed in primary hepatocytes significantly reduced the levels of mitochondrial-specific Mg$^{2+}$ (Mg-S-TTP-AM)[29] and cytosolic-specific Mg$^{2+}$ (Mg-S-AM) (Fig. 3h). In the case of Cnnm4-T495I mutant, increased cytosolic and mitochondrial Mg$^{2+}$ levels were observed in the hepatocytes(Fig. 3h).

Thus silencing of *Cnnm4* reduces mitochondrial ROS and restores Mg$^{2+}$ mitochondrial levels under APAP toxicity.

## Cnnm4 reduces endoplasmic reticulum stress in primary hepatocytes

Cell death is a coordinated phenomenon involving multiple signaling pathways with interplays and crosstalk between ER and mitochondria[42]. To evaluate ER activity[15] in primary hepatocytes under APAP treatment, we examined Ca$^{2+}$ release capacity in the ER by using the Grynkiewicz[43] method and a specific FURA-2 labeling. A lack of *Cnnm4* led to a more functional hepatic ER with a higher release of Ca$^{2+}$ under the different stimuli, including thapsigargin (a specific inhibitor of the ER SERCA pump)[44,45] and ATP (which triggers Ca$^{2+}$ release through P2Y receptor)[46] (Fig. 4a) (Supplementary Fig. 5A).

Regarding the kinases involved in the stress response mediated by APAP treatment, JNK activation has emerged as a crucial event in acute liver injury. A decreasing tendency of activated JNK levels was observed 1 h after APAP administration in si*Cnnm4*-treated hepatocytes compared to controls (Fig. 4b).

Therefore, to investigate whether *Cnnm4* siRNA treatment could reduce ER stress by APAP-induced damage, primary hepatocytes were treated with APAP for a 6 h-time course. The development of ER stress was observed in control cells under APAP treatment, while hepatocytes treated with si*Cnnm4* were significantly protected, as assessed by the mRNA of *Atf6*, *Chop*, *Xbp1*, and *Grp78* (Fig. 4c). In contrast, the overexpression of Cnnm4 affected the levels of genes related to ER stress in the hepatocytes, whereas Cnnm4 T495I did not have the same effect as the WT Cnnm4 (Fig. 4c).

To further identify perturbations in ER function modulated by CNNM4 in the presence of APAP, primary hepatocytes were also labeled with an ER-tracker that binds to the ATP-sensitive K$^+$ channels. Previously data have shown that the ER K$^+$ channel activity is induced under stress[47]. Fluorescence images confirmed the beneficial effect of

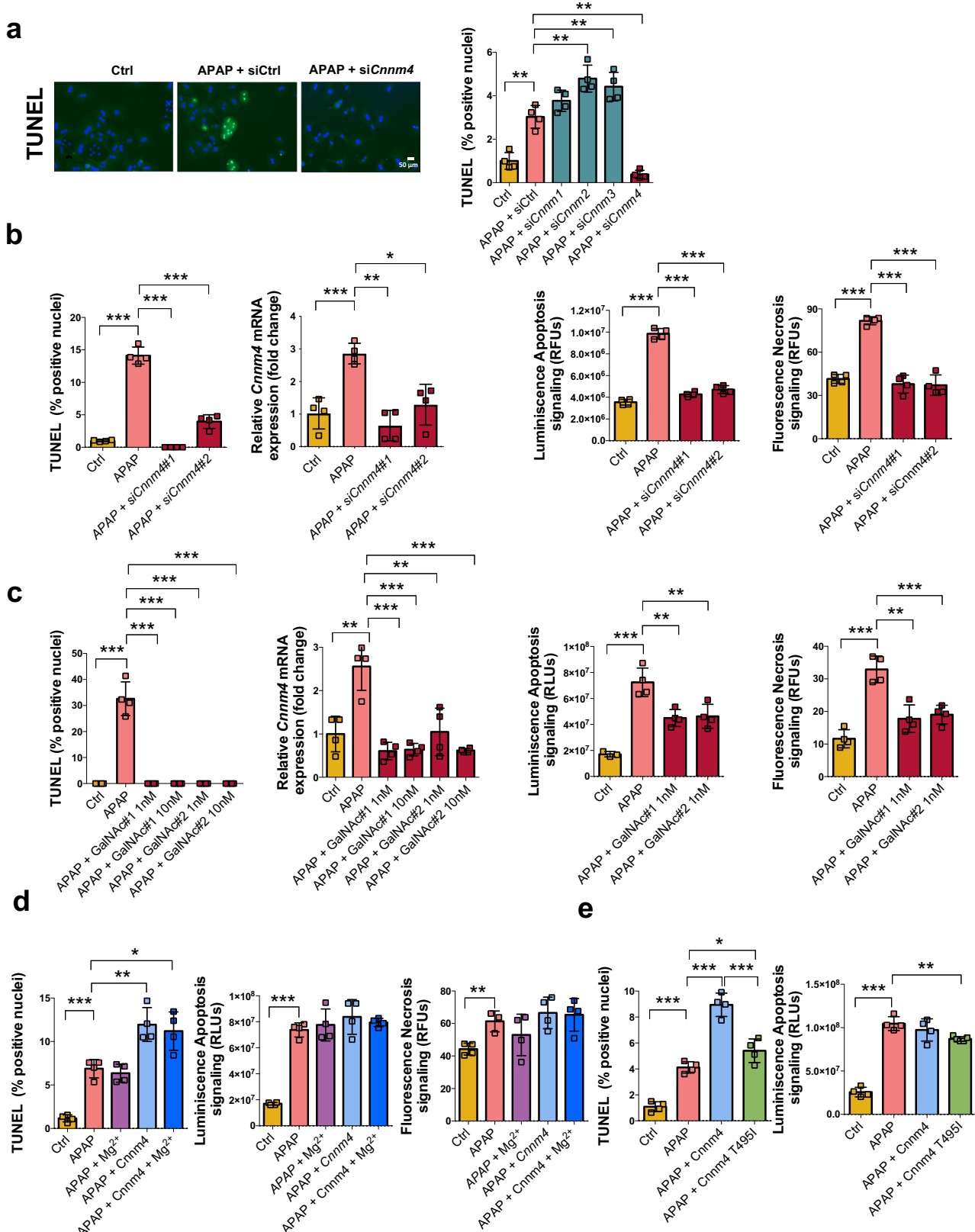

*Cnnm4* knockdown and revealed decreased numbers of red-stained cells under APAP exposure (Fig. 4d). However, Cnnm4 overexpression significantly modulated ER activity, while Cnnm4 T495I did not exert the same effect as the WT Cnnm4 (Fig. 4f).

Finally, hepatocytes treated with a well-known inducer of ER stress, tunicamycin[18], showed a decrease in tunel assay when

*Cnnm4* was knocked down. These results highlight the important role of CNNM4 in hepatocytes under ER stress (Supplementary Fig. 5B).

Thus, we found that *Cnnm4* deficiency alleviates the APAP-mediated ER stress through its $Mg^{2+}$ effluxer activity in primary hepatocytes.

**Fig. 2 | Silencing *Cnnm4* protects against APAP toxicity in *primary and human hepatocytes*. a** Cell death was evaluated using TUNEL in WT hepatocytes under APAP overdose for 3 h and treated with a siRNA *Cnnm1*, *Cnnm2*, *Cnnm3*, and *Cnnm4* or an unrelated control (siCtrl) compared to a control group (Ctrl). Scale bar correspond to 50 μm. Values are represented as mean ± SEM. **$P < 0.01$ ($P = 0.002$ APAP vs Ctrl; $P = 0.005$ APAP + si*Cnnm2* vs APAP; $P = 0.01$ APAP + si*Cnnm3* vs APAP; $P = 0.000074$ APAP + si*Cnnm4* vs APAP) (Student's test, two-sided) **b** Cell death, apoptosis and necrosis reponse were evaluated using TUNEL and Annexin V Apoptosis and necrosis assay respectively in THLE2 cell line under APAP overdose for 3 h and treated with two different siRNA sequences of *Cnnm4* and compared to a control group (Ctrl). Values are represented as mean ± SEM. *$P < 0.05$, **$P < 0.01$, ***$P < 0.001$ (TUNEL $P = 0.00001$ APAP vs Ctrl; $P = 0.00000067$ APAP + si*Cnnm4*#1 vs APAP; $P = 0.000019$ APAP + si*Cnnm4*#2 vs APAP) (mRNA $P = 0.00069$ APAP vs Ctrl; $P = 0.00022$ APAP + si*Cnnm4*#1 vs APAP; $P = 0.0042$ APAP + si*Cnnm4*#2 vs APAP) (Apoptosis $P = 0.00000048$ APAP vs Ctrl; $P = 0.00000002$ APAP + si*Cnnm4*#1 vs APAP; $P = 0.0000028$ APAP + si*Cnnm4*#2 vs APAP) (Necrosis $P = 0.0000012$ APAP vs Ctrl; $P = 0.000014$ APAP + si*Cnnm4*#1 vs APAP; $P = 0.000023$ APAP + si*Cnnm4*#2 vs APAP) (Student's test, two-sided). **c** Cell death, apoptosis, and necrosis reponse were evaluated using TUNEL and Annexin V Apoptosis and necrosis assay respectively were evaluated in primary murine hepatocytes treated with different GalNAc *Cnnm4* siRNAs (GalNAc#1 and GalNAc#2) at different concentrations 1 nM and 10 nM and exposed to APAP for 3 h compared to untreated control group (Ctrl). Values are represented as mean ± SEM. *$P < 0.05$, **$P < 0.01$, ***$P < 0.001$ (TUNEL $P = 0.000055$ APAP vs Ctrl; $P = 0.000055$ APAP + GalNAc#1 1 nM vs APAP; $P = 0.000055$ APAP + GalNAc#1 10 nM vs APAP; $P = 0.000055$ APAP + GalNAc#2 1 nM vs APAP; $P = 0.000055$ APAP + GalNAc#2 10 nM vs APAP) (mRNA $P = 0.0039$ APAP vs Ctrl; $P = 0.00055$ APAP + GalNAc#1 1 nM vs APAP; $P = 0.00051$ APAP + GalNAc#1 10 nM vs APAP; $P = 0.0081$ APAP + GalNAc#2 1 nM vs APAP; $P = 0.00042$ APAP + GalNAc#2 10 nM vs APAP) (Apoptosis $P = 0.0004$ APAP vs Ctrl; $P = 0.005$ APAP + GalNAc#1 1 nM vs APAP; $P = 0.01$ APAP + GalNAc#2 1 nM vs APAP) (Necrosis $P = 0.027$ APAP vs Ctrl; $P = 0.0021$ APAP + GalNAc#1 1 nM vs APAP; $P = 0.0014$ APAP + GalNAc#2 1 nM vs APAP) (Student's test, two-sided). **d** Cell death, apoptosis, and necrosis response were evaluated using TUNEL and Annexin V Apoptosis and necrosis assay in WT hepatocytes under APAP overdose for 3 h with and without 5 mM of $Mg^{2+}$, overexpressing Cnnm4 and overexpressing Cnnm4 with $Mg^{2+}$ compared to a control group (Ctrl). Values are represented as mean ± SEM. *$P < 0.05$, **$P < 0.01$, ***$P < 0.001$ (TUNEL $P = 0.000055$ APAP vs Ctrl; $P = 0.0038$ APAP + Cnnm4 vs APAP; $P = 0.012$ APAP + Cnnm4 + $Mg^{2+}$ vs APAP) (Apoptosis $P = 0.00000088$ APAP vs Ctrl) (Necrosis $P = 0.0062$ APAP vs Ctrl) (Student's test, two-sided). **e** Cell death, apoptosis, and necrosis response were evaluated using TUNEL and Annexin V Apoptosis and necrosis assay in WT hepatocytes under APAP overdose for overexpressing Cnnm4 and overexpressing the mutant Cnnm4 T495I and compared to a control group (Ctrl). Values are represented as mean ± SEM. *$P < 0.05$***$P < 0.001$ (TUNEL $P = 0.000046$ APAP vs Ctrl; $P = 0.00007$ APAP + Cnnm4 vs APAP; $P = 0.044$ APAP + Cnnm4 T495I vs APAP) (Student's test, two-sided). Quadrupled were used for experimental conditions. Source data are provided as a Source Data file.

## Cnnm4 silencing overcomes APAP toxicity in preclinical animal models

Silencing the expression of specific genes by RNA interference has been employed as a therapeutic tool for liver diseases in a number of clinical trials[48]. To determine the protective effect of silencing *Cnnm4* expression, WT mice were treated with a toxic dose of APAP, and 24 h later siRNA *Cnnm4* or an unrelated control (siCtrl) was formulated with Invivofectamine 3.0 and administered by tail vein injection (i.v.). Mice were sacrificed 24 h later and tested for gene expression levels of *Cnnm1-4*. The gene profile performed in the liver of these animals showed that, first of all, *Cnnm4* was the only transporter elevated after APAP-induced injury, and only *Cnnm4* was specifically silenced, although not completely, upon treatment (Fig. 5a and Supplementary Fig. 6A–C). Silencing of *Cnnm4* at the basal level showed no effects, as indicated by hematoxylin and eosin (H&E), alanine transaminase (ALT) and aspartate transaminase (AST) levels, and inflammatory responses (Supplementary Fig. 6D–F), with the exception of serum $Mg^{2+}$ levels (Supplementary Fig. 6G). However, following APAP overdose, H&E staining and TUNEL assay revealed the presence of necrotic areas in the siCtrl-treated cohort caused by the APAP overdose, whereas in the si*Cnnm4*-treated cohort, the liver parenchyma was significantly less disturbed by necrotic areas (Fig. 5b, c). Importantly, the analysis of serum transaminases revealed that si*Cnnm4* treatment significantly reduced their levels (Fig. 5d). To further assess liver injury, we performed DHE staining to detect intracellular superoxide formation in liver sections. Knocking down *Cnnm4* in the liver of APAP-treated mice significantly reduced oxidative stress (Fig. 5e). Accordingly, glutathione species in liver tissues were analyzed. While the GSSG/GSH ratio tended to increase under APAP overdose, the absence of *Cnnm4* reduced this oxidative ratio (Supplementary Fig. 6H).

The inflammatory response is a well-known process during DILI[38]. The analysis of F4/80 staining showed lower numbers of macrophages in Cnnm4-targeted livers (Fig. 5f). This decreased inflammation in the si*Cnnm4* treatment group was further supported by the attenuation of mRNA expression levels of inflammatory cytokines *Tnf*, chemokine (*Cxcl-1*), and chemokine receptor (*Ccl2*) (Supplementary Fig. 6I). The analysis of serum TNF and IL6 determined by ELISA confirmed the rescue of the inflammatory process under *Cnnm4* silencing conditions (Supplementary Fig. 6J). Finally, these data are in agreement with the significantly increased of $Mg^{2+}$ serum level in APAP-treated mice, while the silencing of

*Cnnm4* reduced the $Mg^{2+}$ levels (Fig. 5g). These data confirm the modulation of $Mg^{2+}$ flux by *Cnnm4* silencing in vivo.

To further investigate the beneficial effects of silencing *Cnnm4* specifically in the liver, a GalNAc formulation was used in mice treated with a single dose of APAP 360 mg/kg for 24 h. Animals were sacrificed 48 h after APAP treatment, and CNNM4 levels were assessed by RNA and protein expression, showing significant reductions (Fig. 5h and Supplementary Fig. 7A, B). Necrotic areas, TUNEL assay, and transaminase levels were evaluated (Fig. 5i–k). In addition, the inflammatory response was examined (Fig. 5l). The results show a statistical reduction in liver injury with GalNAc *Cnnm4* siRNA treatment. These data are consistent with the modulation of $Mg^{2+}$ levels in the presence of GalNAc *Cnnm4* siRNA (Fig. 5m). These results bring the treatment of DILI closer to that of GalNAc *Cnnm4* acting specifically in the liver.

N-acetylcysteine (NAC) is the only available therapy for the treatment of patients with APAP overdose, but if administered after 8 h of APAP uptake, its efficacy decreases dramatically[10–13]. To compare the beneficial effects of NAC versus *Cnnm4* silencing, control mice were administered APAP and treated with NAC 24 h later. Then, 48 h after APAP administration, hepatoxicity was examined. In contrast to si*Cnnm4* treatment, liver injury, as determined by histology (Supplementary Fig. 7C), and serum AST and ALT (Supplementary Fig. 7D), were not affected by NAC treatment.

Together, these results show that in this mouse model, applying *Cnnm4* siRNA 24 h after administering an overdose of APAP, a time point when it is non-treatable by NAC, can ameliorate necrosis and inflammatory response in the liver.

It is well established that CNNM4 is a $Mg^{2+}$ transporter highly expressed in the colon epithelia[35]. To determine the liver specificity of *Cnnm4* targeting, CNNM4 levels were measured in the intestine. The expression of CNNM4 increased by APAP treatment in both the control and *Cnnm4* siRNA experimental groups, without significant differences (Supplementary Fig. 8A). Moreover, silencing of *Cnnm4* in healthy mice did not modulate its expression in the intestine, thus excluding non-hepatic sites of action for such silencing (Supplementary Fig. 8B). To further investigate whether liver-specific *Cnnm4* silencing could also be associated with cell differentiation in the colon epithelia, making them more vulnerable to injury, we analyzed intestinal permeability by FITC-dextran. We identified a deficiency in the intestinal integrity of mice treated with APAP, whereas si*Cnnm4* tended to reduce permeability levels closer to normal (Supplementary

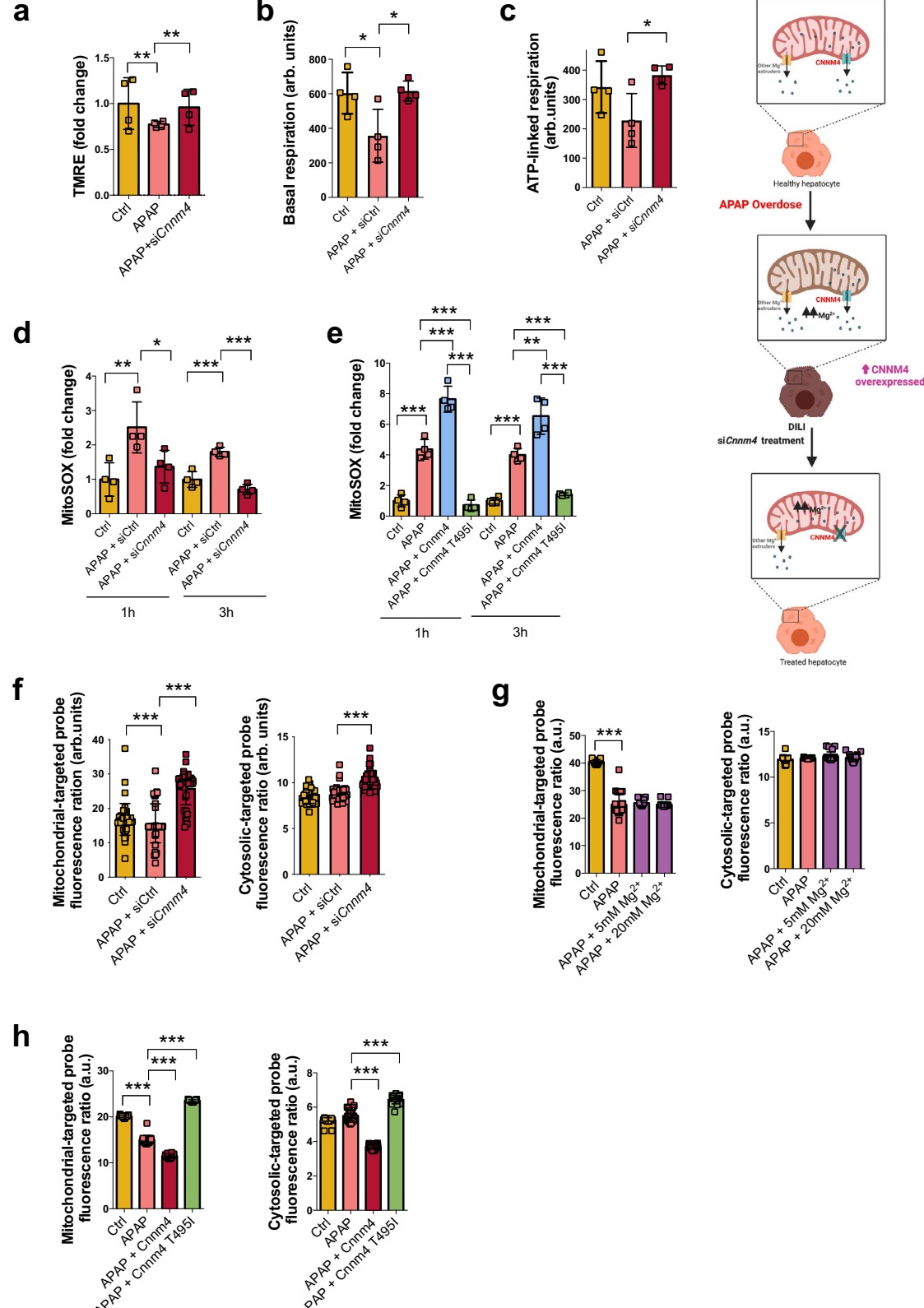

Fig. 8C). Finally, to determine the specificity of CNNM4 IHC staining in the intestine, competitive assays were conducted with the pure protein validating our findings (Supplementary Fig. 8D).

Together these results show that inhibiting *Cnnm4* expression in the liver after APAP administration does not produce any negative effects on the intestinal barrier function.

## Targeting Cnnm4 in preclinical APAP model ameliorates ER stress and mitochondrial ROS through Mg²⁺ modulation

To further interrogate the mechanism underlying *Cnnm4* silencing in DILI, high-throughput proteomics analysis was performed in liver extracts from mice administered an acetaminophen overdose in the presence or absence of this gene and in a control group. Volcano plots

**Fig. 3 | Silencing *Cnnm4* in hepatocytes reduces mitochondrial dysfunction under APAP toxicity. a** The mitochondrial membrane potential was determined in WT hepatocytes upon 3 h of APAP overdose and treated with a siRNA *Cnnm4* or an unrelated control. Values are represented as mean ± SEM. \**$P < 0.01$, \***$P < 0.001$ ($P = 0.011$ APAP vs Ctrl; $P = 0.0037$ APAP + si*Cnnm4* vs APAP) (Student's test, two-sided). **b** Basal respiration using seahorse assay in primary hepatocytes and after 3 h of APAP administration. Values are represented as mean ± SEM. \*$P < 0.05$ ($P = 0.043$ APAP vs Ctrl; $P = 0.02$ APAP + si*Cnnm4* vs APAP) (Student's test, two-sided). **c** Mitochondrial ATP production and linked respiration in primary hepatocytes under APAP overdose for 3 h. Values are represented as mean ± SEM. \*$P < 0.05$, \**$P < 0.01$, \***$P < 0.001$ ($P = 0.019$ APAP + si*Cnnm4* vs APAP) (Student's test, two-sided). The mitochondrial Reactive Oxygen Species was determined in WT hepatocytes upon 1 h and 3 h of APAP treatment using MitoSOX staining and treated with **d.** siRNA *Cnnm4* or an unrelated control. Values are represented as mean ± SEM. \*$P < 0.05$, \**$P < 0.01$, \***$P < 0.001$ (1 h $P = 0.01$ APAP vs Ctrl; $P = 0.04$ APAP + si*Cnnm4* vs APAP) (3 h $P = 0.00089$ APAP vs Ctrl; $P = 0.000033$ APAP + si*Cnnm4* vs APAP) and **e** Cnnm4 and Cnnm4 T495I overexpression. Values are represented as mean ± SEM. \*$P < 0.05$, \**$P < 0.01$, \***$P < 0.001$ (1 h $P = 0.0001$ APAP vs Ctrl;

$P = 0.00082$ APAP + Cnnm4 vs APAP; $P = 0.0000056$ APAP + Cnnm4 T495I vs APAP) (3 h $P = 0.000014$ APAP vs Ctrl; $P = 0.0069$ APAP + Cnnm4 vs APAP; $P = 0.00002$ APAP + Cnnm4 T495I vs APAP) (Student's test, two-sided). Relative intracellular $Mg^{2+}$ determination by mitochondrial-specific labeling and cytosolic-specific labeling in primary hepatocytes under APAP overdose for 3 h and treated with **f.** a siRNA *Cnnm4* or an unrelated control (siCtrl) compared to a control group (Ctrl). Values are represented as mean ± SEM. \***$P < 0.001$ (Mitochondrial $P = 0.0000000000004$ APAP vs Ctrl; $P = 0.000000000000026$ APAP + si*Cnnm4* vs APAP) (Cytosolic $P = 0.0000000044$ APAP + si*Cnnm4* vs APAP) (Student's test) **g** 5 mM and 20 mM $Mg^{2+}$ supplementation. Values are represented as mean ± SEM. \***$P < 0.001$ (Mitochondrial $P = 0.0000000000000059$ APAP vs Ctrl) (Student's test, two-sided). **h** Cnnm4 and Cnnm4 T495I overexpression compared to a control group (Ctrl). Quadrupled were used for experimental condition. Values are represented as mean ± SEM. \***$P < 0.001$ (Mitochondrial $P = 1.2E-39$ APAP vs Ctrl; $P = 2.5E-08$ APAP + Cnnm4 + Cnnm4 WT vs APAP; $P = 3.61E-45$ APAP + Cnnm4 T495I vs APAP) (Cytosolic $P = 2.18E-24$ APAP + Cnnm4 WT vs APAP; $P = 8.15E-22$ APAP + Cnnm4 T495I vs APAP) (Student's test, two-sided). Source data are provided as a Source Data file. The image was created with BioRender.com.

show the more representative proteins modulated in the absence of CNNM4 (Fig. 6a). For more specific detail, the top-50 up- and down-regulated proteins linked to CNNM4 presence under APAP treatement were represented in a heatmap (Supplementary Fig. 9A). The Database for Annotation, Visualization, and Integrated Discovery (DAVID) was used to analyze the major signaling pathways altered by *Cnnm4* knockdown (Fig. 6b). Processes downregulated by the absence of *Cnnm4* included those involved in mitochondrial membrane and ATP metabolic pathways, and in particularly gene ontology families related to ER activity, protein transport, misfolding, and their catabolic regulation (Fig. 6b). In contrast, pathways involved in the regulation of cytokine-mediated signaling; interferon alpha, implicated in liver regeneration, among others processes; and the response to interleukin 10 appeared to be overrepresented in these circumstances (Fig. 6b).

Based on these data, first we investigated whether reduction of *Cnnm4* expression in vivo protects against liver injury by modulating the three major pathways mediated by the ER-resident transmembrane proteins ATF6, IRE1, and PERK. We found that a lack of *Cnnm4* attenuated the translation of the splicing of *Xbp1* mRNA, related to IRE1 activation (Fig. 6c). These results are in accordance with the observed reduction of *Chop* (Fig. 6c). Moreover, we observed a decreasing tendency of *Atf6* and *Grp78* levels silencing *Cnnm4* (Fig. 6c). Accordingly, APAP stimulated the phosphorylation of EIF2alpha, which was attenuated by si*Cnnm4* (Fig. 6d).

Moreover, the GalNAc *Cnnm4* siRNA therapeutic approach under APAP treatment in vivo also led to a reduction in ER stress, as shown in Fig. 6e, f.

These results suggest that *Cnnm4* silencing treatment restores normal $Mg^{2+}$ homeostasis and reduces the ER stress triggered by APAP overdose in in vivo animal models of APAP-induced liver failure.

To better understand the effects mediated by CNNM4 on ER activity under APAP exposure, the cellular localization of this protein was analyzed in the liver of DILI animal models (Fig. 6g). Highly enriched fractions from liver tissue allowed us to monitor the downregulation of CNNM4 levels in the membrane fraction in the presence of APAP compared with increased ER under these conditions (Fig. 6g). The mitochondrial fraction did not exhibit CNNM4.

Finally, considering the interplay between ER and mitochondria and the disturbance of $Mg^{2+}$ content in the mitochondria under *Cnnm4* silencing, mitochondrial respiration was evaluated in mitochondria from APAP-treated mice with GalNAc si*Cnnm4* and unrelated control (Fig. 6h). ROS production and ATP levels were evaluated in hepatocytes derived from APAP-treated mice and those under GalNAc *Cnnm4* treatment (Fig. 6i, j). The data identified that GalNAc si*Cnnm4* also improved mitochondrial activity and its bioenergetics. In this context, a significantly reduction in ER tracker was also identified (Fig. 6k).

## CNNM4 knockdown induced liver regeneration in preclinical APAP model

Increased ATP levels in the liver after injury could facilitate liver regeneration, an energetically demanding process[12]. To further assess the impact of ATP production under *Cnnm4* silencing in liver regeneration, the proliferative response was evaluated in mice subjected to liver damage 48 h after APAP overdose. Analysis of cyclin D1 and A2 gene expression in liver tissue showed increased levels in si*Cnnm4*-treated mice (Fig. 7a). In addition, *Cnnm4* silencing induced proliferating cell nuclear antigen (PCNA) staining in liver sections (Fig. 7b). Further analysis of the upstream regulators mediating liver regeneration after APAP injury revealed that hepatocyte growth factor (HGF), considered critical for hepatocyte proliferation[49], was significantly increased under these conditions (Fig. 7c). Liver regeneration evaluated at an early time, 36 h for mice administered APAP, and 12 h later of GalNAc *Cnnm4* injection resulted (Fig. 7d) in a reduction of p21 cell cycle arrest (Fig. 7e) and significant activation of phospho c-Met (Fig. 7f). Downregulation of the inflammatory response mediated by F4/80 was identified under GalNAc *Cnnm4* silencing (Supplementary Fig. 10A–E)

Thus, targeting *Cnnm4* at late phases of APAP toxicity can induce a regenerative response in the liver, thus counteracting the damage produced after the administration of this drug.

## Discussion

DILI is a clinical and pathological term for liver injury caused by various medications, leading to abnormalities in liver tests or liver dysfunction. DILI is one of the leading causes of acute liver failure, which can result in the need for liver transplantation or even in death[50]. APAP overdose is one of the principal causes of DILI[51], with necrosis as the primary pathway of hepatocyte cell death. Understanding the mechanisms behind APAP-mediated toxicity is therefore essential in order to overcome liver damage with appropriate therapies. In this work, we took the challenge of identifying $Mg^{2+}$ perturbations in DILI patients and studied the impact by modulating $Mg^{2+}$ in preclinical AILI models. Our findings reveal that CNNM4 is a major mediator of $Mg^{2+}$ efflux from liver to extracellular compartments. We show that *Cnnm4* silencing under APAP exposure results in the restoration of $Mg^{2+}$ levels within the mitochondria, ameliorating ER stress and protecting mitochondria functionality (Fig. 8). Our observation that hepatic CNNM4 is upregulated in the preclinical animal models as well as in ALF patients who underwent liver transplant due to acetaminophen overdose supports this premise. Based on our findings, a therapeutic window of 6 to 24 h following acetaminophen overdose ingestion appears suitable. Nevertheless, further research will help to determine whether broader therapeutic frames may be applied.

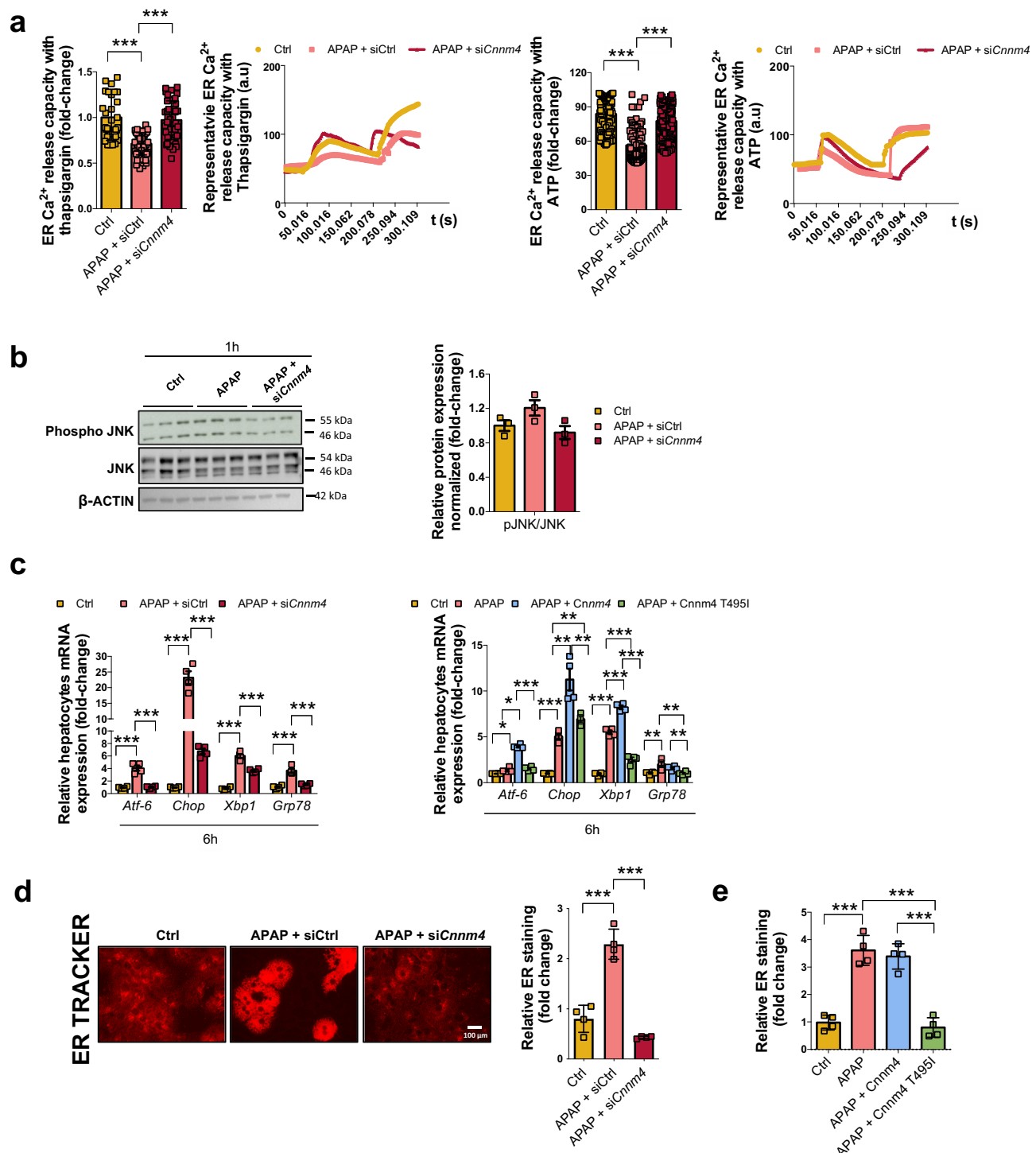

Mg$^{2+}$ is essential for health and plays a role in virtually every process within the human cell. Given such important functions, it is clear that perturbations in intracellular Mg$^{2+}$ concentrations could have serious implications for the physiological functioning of cells. Dysfunction of Mg$^{2+}$ handling is linked to different pathologies, including liver diseases[5], diabetes[3], hypertension[3], and cardiovascular diseases. Here, we show that CNNM4 levels are elevated in primary hepatocytes challenged with APAP in the liver of mice after APAP overdose, and in liver tissue from DILI patients with ALF. Thus, APAP appears to trigger Cnnm4 overexpression in the liver, with a concomitant Mg$^{2+}$ efflux from the hepatocytes to the extracellular media. Consistent with this, serum Mg$^{2+}$ levels both in the APAP overdose

animal model and in APAP DILI patients were increased. Importantly, under conditions of liver toxicity, CNNM4 appears to accumulate in the ER[52]. Indeed, an analysis by the SubCons webserver identified a signal peptide responsible for the ER localization of CNNM4. High-throughput proteomics analysis revealed that in the absence of *Cnnm4*, pathways related to ER activity, protein transport, misfolding, and their catabolic regulation was mainly affected under APAP overdose. CNNM4 upregulation in the ER may explain the Mg$^{2+}$ disturbances observed in the hepatocytes in ALF.

The pattern of Mg$^{2+}$ levels in primary hepatocytes, determined by mitochondrial-specific (Mg-S-TTP-AM)[29] and cytosolic-specific (Mg-S-AM)[41] labeling, showed that CNNM4 acted as an effluxer reducing Mg$^{2+}$

**Fig. 4 | Silencing *Cnnm4* in hepatocytes reduces endoplasmic reticulum stress under APAP toxicity. a** Calcium (Ca$^{2+}$) release capacity by ER with thapsigargin and ATP upon time under APAP overdose for 3 h treated with a siRNA *Cnnm4* or an unrelated control (siCtrl) compared to a healthy group (Ctrl). Triplicates were used for experimental condition. Values are represented as mean ± SEM. ***$P < 0.001$ (Thapsigargin $P = 6E-08$ APAP vs Ctrl; $P = 1.23E-07$ APAP + si*Cnnm4* vs APAP) (ATP $P = 1.3E-10$ APAP vs Ctrl; $P = 1.24E-11$ APAP + si*Cnnm4* vs APAP) (Student's test, two-sided). **b** Western blot analysis of phospho-JNK1/JNK2 (Thr183, Tyr185) (pJNK) and SAPK/JNK (Thr183, Tyr185) ratio, β-ACTIN was used as a loading control under APAP overdose for 1 h. **c** mRNA levels of *Atf-6, Chop, X-box binding protein 1 (Xbp1s/Xbp1u)* and binding immunoglobulin protein (*Grp78*) for 6 h under APAP overdose and treated with **a** a siRNA *Cnnm4* or an unrelated control (siCtrl) compared to a control group (Ctrl). **b** Cnnm4 and Cnnm4 T495I overexpression compared to a control group (Ctrl). Values are represented as mean ± SEM. ***$P < 0.001$ (Atf6 $P = 0.0003$ APAP vs Ctrl; $P = 0.00035$ APAP + si*Cnnm4* vs Ctrl) (Chop $P = 3.83E-05$ APAP vs Ctrl; $P = 0.00024$ APAP + si*Cnnm4* vs Ctrl) (Xbp1 $P = 4.95E-06$ APAP vs Ctrl; $P = 0.00063$ APAP + si*Cnnm4* vs APAP) (Grp78 $P = 0.0003$ APAP vs Ctrl; $P = 0.00057$ APAP + si*Cnnm4* vs APAP) *$P < 0.05$; **$P < 0.01$; ***$P < 0.001$ (Atf6 $P = 0.03$ APAP vs Ctrl; $P = 4.64E-06$ APAP + Cnnm4 vs APAP) (Chop $P = 8.38E-06$ APAP vs Ctrl; $P = 0.0022$ APAP + Cnnm4 vs APAP; $P = 0.0037$ APAP + Cnnm4 T495I vs APAP) (Xbp1 $P = 7.96E-07$ APAP vs Ctrl; $P = 8.17E-05$ APAP + Cnnm4 vs APAP; $P = 8.19E-05$ APAP + Cnnm4 T495I vs APAP) (Grp78 $P = 0.016$ APAP vs Ctrl; $P = 0.015$ APAP + Cnnm4 T495I vs APAP) (Student's test, two-sided). **d** ER tracker red staining with respective quantification in primary hepatocytes for 3 h under APAP overdose and treated with a siRNA *Cnnm4* or an unrelated control (siCtrl) compared to a control group (Ctrl). Values are represented as mean ± SEM. ***$P < 0.001$ ($P = 0.00032$ APAP vs Ctrl; $P = 1.73E-05$ APAP + si*Cnnm4* vs APAP) (Student's test, two-sided) and **e** ER tracker red staining with respective quantification in primary hepatocytes for 3 h under APAP overdose and treated with Cnnm4 and Cnnm4 T495I overexpression. Scale bar correspond to 100 μm. Quadrupled were used for experimental condition. Values are represented as mean ± SEM. ***$P < 0.001$ ($P = 0.00013$ APAP vs Ctrl; $P = 0.00013$ APAP + Cnnm4 T495I vs APAP) (Student's test, two-sided). Source data are provided as a Source Data file.

levels in the cell and in the mitochondria. There are several signaling mechanisms linking ER stress and mitochondrial cell death pathways. MAMs are the interaction zones between the two organelles that allow rapid exchange of molecules; among them, Ca$^{2+}$ maintains cellular function and cellular health. Thus, the presence of CNNM4 in the ER under APAP toxicity probably dysregulates ER Ca$^{2+}$ homeostasis and induces a misfolding protein response and mitochondrial ROS. The overexpression of Cnnm4 and the results obtained in the presence of the inactive mutant T495I[30] unable to induce Mg$^{2+}$ efflux support the role of CNNM4 in the ER. Indeed, targeting *Cnnm4* in hepatocytes under APAP treatment preserves ER activity and reduces the stress mediated by this toxicant ameliorating cell death. This effect appears to be dependent on the cells in which the modulation of CNNM4 occurs. Previous work has shown that silencing of *Cnnm4* in intestinal epithelial cells triggers an oxidative stress response[53]. Hepatocytes, which are rich in mitochondria, have a high oxidative capacity[54]. This characteristic probably leads to a differential response under *Cnnm4* knockdown. Therefore, *Cnnm4* should specifically be silenced in certain cell types, as is the case with GalNAc si*Cnnm4* in the hepatocytes.

Mg$^{2+}$ deficiency has been previously reported in patients with liver disease, although the relationship between this cation, liver function, and the disease has not been fully elucidated[52]. CNNM4, which is upregulated in NAFLD patients, has been described as a master regulator of VLDL export in this pathology, mediated by activation of the microsomal transfer protein found in the ER[29]. Under steatotic conditions[29] or in primary hepatocytes where Cnnm4 was overexpressed or treated with APAP, Mg$^{2+}$ supplementation did not result in any recovery effect or protection from the necrotic processes. Then, hepatic *Cnnm4* silencing appeared to be required for amelioration of Mg$^{2+}$ disturbances in liver disease due to its efflux activity. In this context, it is important to highlight that common to the two pathologies, NAFLD and ALF, is that that reticulum stress is one of the main altered pathways through which CNNM4 can exert its therapeutic effects.

Liver regeneration plays a critical role in resolving APAP-induced ALF[49]. Cellular ATP levels are essential for liver regeneration[55]. Hepatic cell death triggered by APAP is closely related to cellular ATP deprivation caused mainly by mitochondrial dysfunction. In the absence of *Cnnm4*, ATP levels increase, favoring proliferation and preventing hepatic necrosis, which is mainly due to decreased mitochondrial ROS and reticulum stress. Additionally, *Cnnm4* deficiency in APAP injury leads to upregulation of HGF and consequent phosphorylation of c-MET. HGF has been described to play a fundamental role in regeneration after ALF[49]. Moreover, the absence of *Cnnm4* under these experimental conditions also leads to decreased p21 levels. p21 has been directly linked with the impairment liver regeneration observed in patients after severe APAP-induced liver injury in patients[56]. This reduction in p21 along with increased cyclin D1 and PCNA levels suggests that the absence of *Cnnm4* enhances the regenerative response in ALF.

NAC is the only currently available effective therapy that can be deacetylated to cysteine and subsequently metabolized to GSH[53,57], although it has a limited therapeutic window. Thus, more effective therapeutic interventions are urgently needed. Our data have shown that at the time *Cnnm4* silencing appears to have therapeutic value in an ALF animal model, NAC is not a treatment option.

Overall, our results identified that CNNM4 appears to be a major effluxer responsible for Mg$^{2+}$ dysregulation in ALF in the liver. The enrichment of CNNM4 in the ER may justify the stress response and mitochondrial dysfunction observed in in vitro and in vivo models of APAP-mediated hepatotoxicity. Our data from treated mice demonstrate that *Cnnm4* inhibition within 24 h of APAP overdose represents a potential therapeutic avenue for APAP-induced liver injury, with additional benefits for liver regeneration and recovery from injury.

## Methods
### Human samples
All trial studies in this research comply with all relevant ethical regulations. All human study participants involved in these studies have signed a written, informed consent and they have not received any participant's compensation.

**Cohort 1.** Serum biological samples from 4 control individuals with no liver diseases and tolerant to commonly used drugs attending routine work monitoring were used to measure magnesium levels (Hospital Virgen de la Victoria, Málaga, Spain). The study was approved by the Local Ethical Committee (FIM-HEP-2016-01) at the Virgen de la Victoria Hospital in Málaga, Spain.

**Cohort 2.** Twenty-four serum samples from patients with acute liver disease after a single acute APAP overdose with known time of drug ingestion were used to determine magnesium levels. A detailed characterization of the patients is summarized in Supplementary Table I (The Queen's Medical Research Institute). Primary outcomes of a randomized open-label exploratory, safety, and tolerability study with calmangafodipir in patients treated with a 12 h regimen of N-acetylcysteine for paracetamol overdose (POP trial) were previously reported[58]. This study, EudraCT number 2017-000246-21 and ClinicalTrials.gov identifier NCT03177395, was approved by the UK medicines regulator, MHRA (25th April 2017) and West of Scotland Research Ethics Committee 1, Glasgow, UK (11th April 2017).

**Cohort 3.** A total of 13 samples from explants of patients undergoing urgent liver transplantation for ALF resulting from acetaminophen

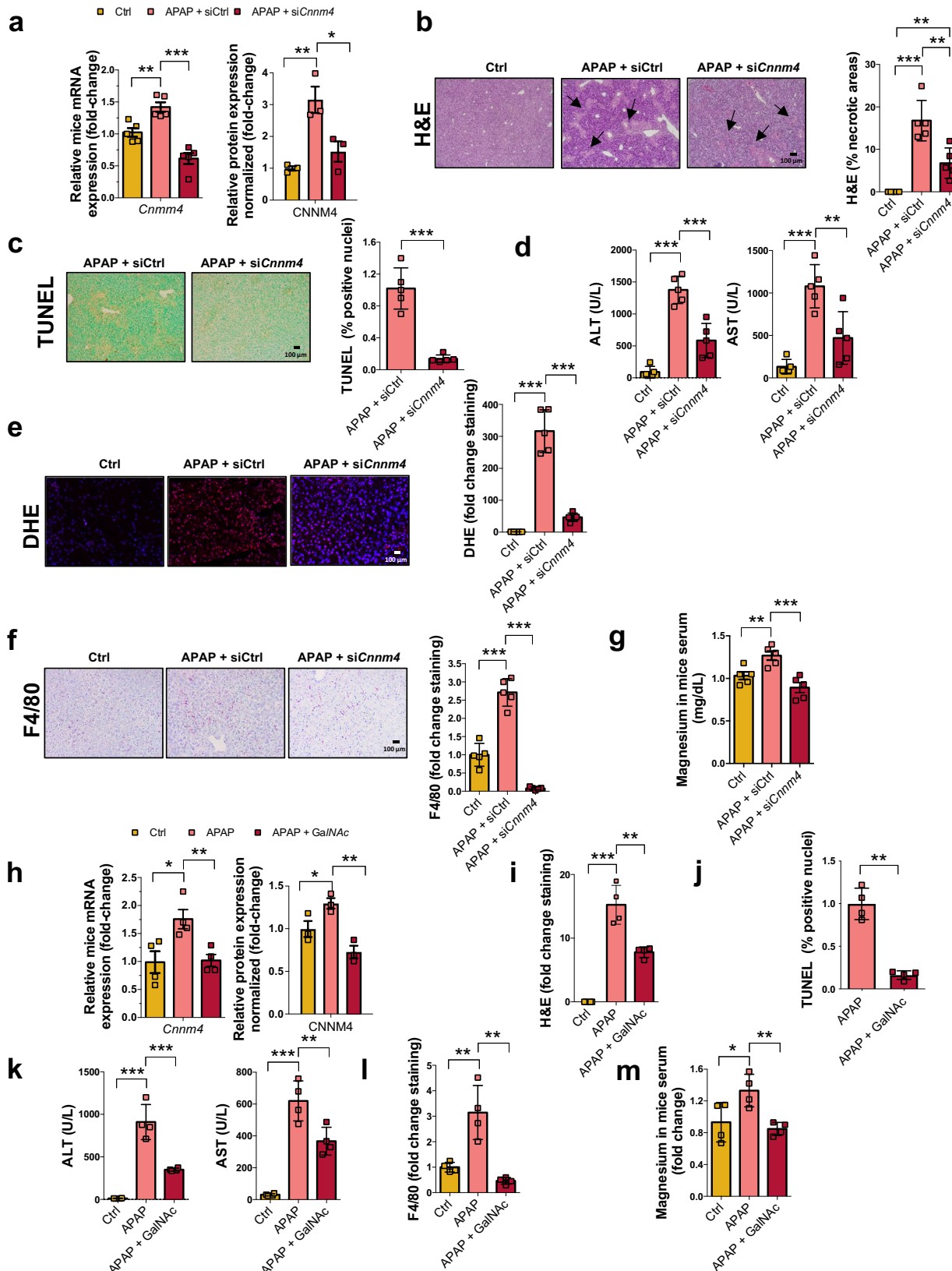

overdose were evaluated for CNNM4 expression. The diagnosis was established by the Newcastle Hospitals NHS Foundation Trust (Newcastle, England) based on clinical data. Detailed descriptions of the patients are given in Supplementary Table II. The timing between overdose and the explant for the patients with hyper-acute liver failure (from paracetamol) was usually 3 days. Newcastle patient tissues were

shared from the Newcastle biomedicine biobank (12/HE/0395), approved by the Newcastle Research Ethics Committee of North East Newcastle and North Tyneside.

**Cohort 4.** Finally, three samples from healthy organ transplant donors without liver lesions or obesity (Marqués de Valdecilla University

**Fig. 5 | Silencing *Cnnm4* prevents hepatotoxicity mediated by APAP overdose in in vivo models.** WT mice were treated with a single dose of APAP 360 mg/kg by intraperitoneal injection and 24 h after siRNA *Cnnm4* ($n = 5$) or an unrelated control (siCtrl) ($n = 5$) were injected via tail vein injection and compared to control group ($n = 5$). **a** mRNA levels and protein expression levels of CNNM4. Values are represented as mean ± SEM. *$P < 0.05$; ***$P < 0.001$ (mRNA $P = 0.0042$ APAP vs Ctrl; $P = 9.98E-05$ APAP + si*Cnnm4* vs APAP) (protein $P = 0.0068$ APAP vs Ctrl; $P = 0.036$ APAP + si*Cnnm4* vs APAP) (Student's test, two-sided). **b** Liver necrosis was assessed by H&E staining. Scale bar correspond to 100 μm. Values are represented as mean ± SEM. **$P < 0.01$; ***$P < 0.001$ ($P = 4.88E-05$ APAP vs Ctrl; $P = 0.0055$ APAP + si*Cnnm4* vs APAP) (Student's test, two-sided). **c** Cell death was determined by TUNEL assay in liver tissue. Scale bar correspond to 100 μm. Values are represented as mean ± SEM. ***$P < 0.001$ ($P = 7.37E-05$ APAP + si*Cnnm4* vs APAP) (Student's test, two-sided). **d** Transaminases ALT and AST levels were determined in mice serum. Values are represented as mean ± SEM. ***$P < 0.001$ (ALT $P = 1.61E-06$ APAP vs Ctrl; $P = 0.00084$ APAP + si*Cnnm4* vs APAP) (AST $P = 4.9E-05$ APAP vs Ctrl; $P = 0.0095$ APAP + si*Cnnm4* vs APAP) (Student's test, two-sided). **e** ROS in vivo measured by DHE staining in liver sections. Scale bar correspond to 100 μm. Values are represented as mean ± SEM. ***$P < 0.001$ ($P = 5.61E-06$ APAP vs Ctrl; $P = 2.06E-07$ APAP + si*Cnnm4* vs APAP) (Student's test, two-sided). **f** Inflammation assessed by F4/80 staining in liver. Scale bar correspond to 100 μm. Values are represented as mean ± SEM. ***$P < 0.001$ ($P = 5.28E-05$ APAP vs Ctrl; $P = 3.02E-07$ APAP + si*Cnnm4* vs APAP) (Student's test, two-sided). **g** Magnesium levels were determined in mice serum ($n = 5$). Values are represented as mean ± SEM. **$P > 0.01$; ***$P < 0.001$ ($P = 0.0087$ APAP vs Ctrl; $P = 0.001$ APAP + si*Cnnm4* vs APAP) (Student's test, two-sided). WT mice were treated with a single dose of APAP 360 mg/kg by intraperitoneal injection and 24 h thereafter with 3 mg/kg GalNAc *Cnnm4* siRNA ($n = 4$) or a nontargeting GalNAc conjugated control siRNA ($n = 4$) were injected via subcutaneous injection. Mice were sacrificed 48 h after APAP overdose. **h** mRNA levels and protein expression of CNNM4 was determined. Values are represented as mean ± SEM. *$P < 0.05$; **$P < 0.001$ (mRNA $P = 0.02$ APAP + GalNAc vs APAP) (protein $P = 0.055$ APAP vs Ctrl; $P = 0.004$ APAP + GalNAc vs APAP) (Student's test, two-sided). **i** Liver necrosis was assessed by H&E staining at 48 h after APAP overdose. Values are represented as mean ± SEM. **$P < 0.01$; ***$P < 0.001$ ($P = 0.0004$ APAP vs Ctrl; $P = 0.01$ APAP + GalNAc vs APAP) (Student's test, two-sided). **j** Cell death was assessed by TUNEL assay. Values are represented as mean ± SEM. **$P < 0.001$ ($P = 0.0031$ APAP + GalNAc vs APAP) (Student's test, two-sided). **k** Transaminases ALT and AST levels were determined in mice serum at 48 h after APAP overdose. Values are represented as mean ± SEM. **$P < 0.01$; ***$P < 0.001$ (ALT $P = 0.0001$ APAP vs Ctrl; $P = 0.0017$ APAP + GalNAc vs APAP) (AST $P = 8.42E-05$ APAP vs Ctrl; $P = 0.01$ APAP + GalNAc vs APAP) (Student's test, two-sided). **l** Inflammation assessed by F4/80 staining in liver at 48 h after APAP overdose. Values are represented as mean ± SEM. ***$P < 0.001$ ($P = 0.0073$ APAP vs Ctrl; $P = 0.0025$ APAP + GalNAc vs APAP (Student's test, two-sided). **m** Magnesium levels were determined in mice serum at 48 h of APAP overdose. Respective quantifications are represented as mean ± SEM. *$P < 0.05$, **$P < 0.01$ ($P = 0.04$ APAP vs Ctrl; $P = 0.0038$ APAP + GalNAc vs APAP (Student's test, two-sided). Source data are provided as a Source Data file.

---

Hospital, Santander, Spain), were used as controls for immunostaining analyses. The study was approved by the Research Ethics Committee of IDIVAL Cantabria 2017.052.

## Animal model for acetaminophen-induced liver damage

All animal experimentation was carried out in accordance with Spanish Guide for the Care and use of Laboratory animals, and with International Animal Care and Use of Committee. Male adult (3-month-old) C57BL/6 mice were acquired from Charles River (St Germain sur l'Arbresle, France) and maintained at CIC bioGUNE animal facility. Animal procedures were approved by the CIC bioGUNE Institutional Animal Care and Use Committee and the local authority (Diputación de Bizkaia).

Mice were kept in a temperature-controlled animal facility (AAA-LAC-accredited) with 12-h light/dark cycles, and fed a standard diet (Harlan Tekland) with water ad libitum.

Liver samples were harvested, snap-frozen in liquid nitrogen, and stored at −80 °C for subsequent analysis.

Mice were starved for 12 h and then given acetaminophen (APAP, Sigma Aldrich) by intraperitoneal injection in a single dose of 360 mg/kg. Animals were sacrificed at 24, 36, and 48 h after APAP overdose.

## GalNAc siRNAs

GalNAc#1 and #2 comprise double-stranded 19mer RNAs with 2′-O-methyl, 2′-fluoro-2′deoxy modifications and phrophorothioate bonds targeting *Cnnm4* linked to triantennary GalNAc unit. Non-targeting control siRNA conjugate represents a siRNA for Luciferase (Photinus pyralis) linked to the same GalNAc unit. *Cnnm4* molecules were designed, synthesized, and purified by Silence Therapeutics GmbH, Berlin[59].

## In vivo silencing

After 24 h of APAP treatment, mice were randomly divided into two experimental groups ($n = 4$), 27 μM of silenced *Cnnm4* (si*Cnnm4*) or unrelated siRNA control (siCtrl), (Custom Ambion® In Vivo siRNA, Cat#: 4457302) by intravenous tail injection using Invivofectamine 3.0 Reagent (Thermo Fisher Scientific). Mice were sacrificed 48 h after APAP overdose.

Mice were also silenced employing *Cnnm4* GalNAc siRNA molecule. Thus, after 24 h of APAP treatment, mice were randomly divided into two experimental groups ($n = 4$), GalNAc si*Cnnm4* or control

GalNAc siRNA were administered by subcutaneous injection. Mice were sacrificed at 36 and 48 h after APAP overdose.

## Animal treatment

After mice were starved for 12 h and received a single dose of 360 mg/kg of acetaminophen for 24 h, mice were treated with a single 200 μl intraperitoneal injection of 1200 mg/kg N-acetylcysteine (NAC, Sigma Aldrich). Animals were sacrificed at 48 h after APAP administration.

## Blood collection

Blood samples were obtained by puncture with a heparinized capillary tube[60] in each animal via retroorbital plexus technique. All samples were deposited in serum separator gel tubes (Microtainer, Becton–Dickinson) and centrifuged (2500 *g*, 5 min, 4 °C) for serum separation. Serum was transferred to a new tube and stored at −80 °C. Transaminases and biochemistry blood were analyzed using a Selectra Junior Spinlab 100 analyser (Vital Scientific) according to manufacturers' protocol.

## Extracellular magnesium quantification

QuantiCrom™ Magnesium Assay Kit (BioAssay Systems) was used for extracellular Mg²⁺ determination according to the manufacturer's instructions. $Mg^{2+}$ concentrations were calculated by comparing the OD500 from a standard concentration (2 mg/ml).

## Determination of intestinal permeability in vivo

Four hours before sacrificed, in mice treated with 360 mg/kg of APAP and under si*Cnnm4* or SiCtrl. i.v fluorescein isothiocyanate-dextran (FITC dextran, Sigma) was applied orally at a single dose (0,6 mg of dextran per 23 g of animal weight). Serum samples were collected and fluorescence was determined at 485 nm excitation and 528 nm of emission.

## Histology

*Immunohistochemistry:* Paraffin-embedded liver and intestine samples were sectioned, dewaxed and hydrated. CNNM4 staining was performed according to standard protocols using 1/100 dilution of CNNM4 antibody (Proteintech, 14066-1-AP) and incubating in 2% BSA in 0.01% PBS-azide for 1 h at room temperature. *H&E staining:* Paraffin-embedded liver samples were stained for hematoxylin and eosin

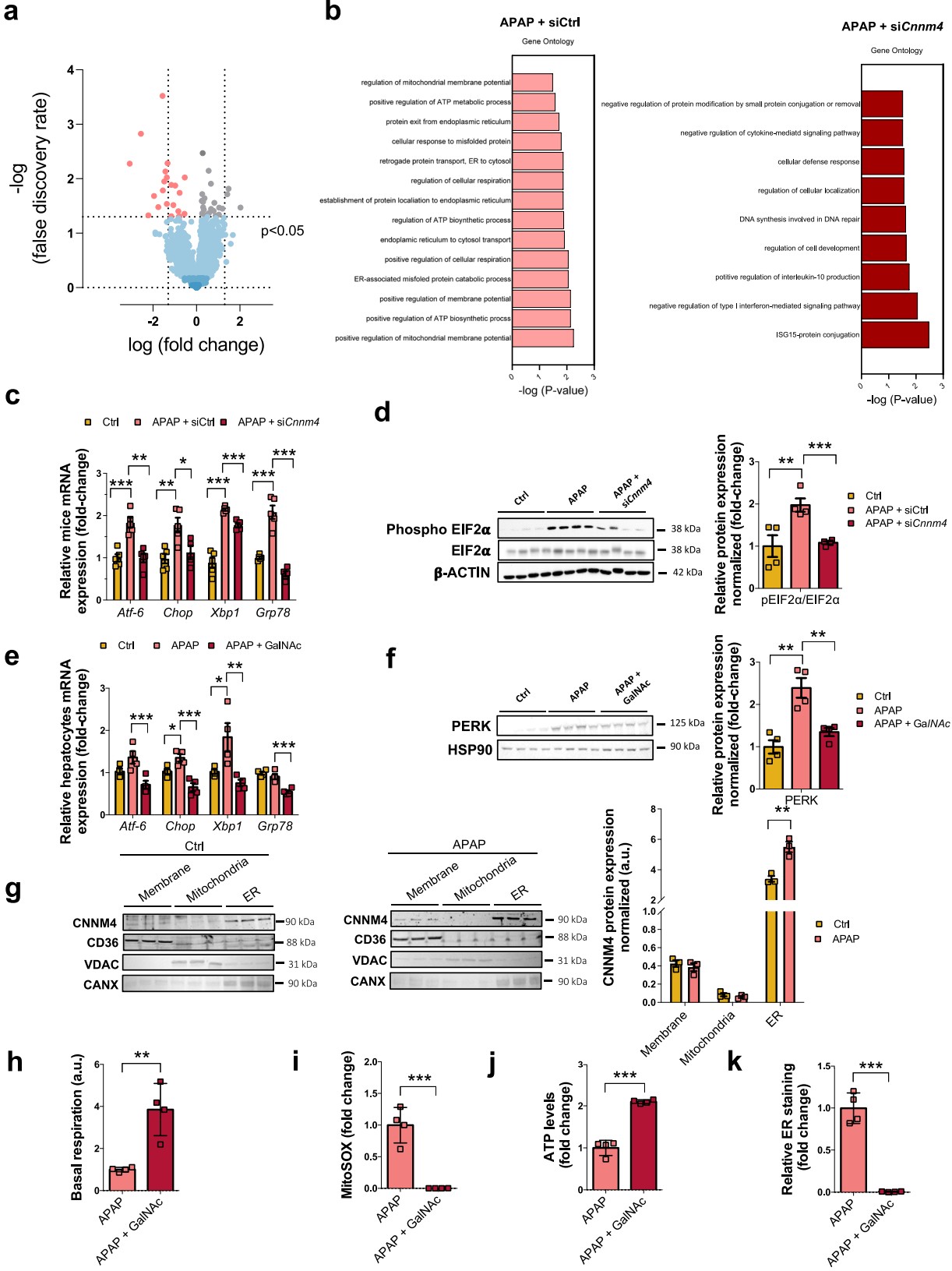

(H&E). *F4/80* was determined as macrophage marker membrane. All procedures were performed according to standard protocols using the EnVision+ System HRP (Dako, Denmark). Samples were incubated with Vector Vip substrate (Vectorlabs, Burlingame, USA) for color development. Five to ten random images per sample were taken with an AXIO Imager A1 microscope (Carl Zeiss AG, Jena, Germany). Stained area percentage of each sample were calculated using FIJI (ImageJ). *Tunel assay* was determined as cell death marker. All procedures were performed according to manufacturer's protocol using Tunel Assay kit-HRP-DAB (ab206386, abcam)

**Fig. 6 | Silencing Cnnm4 reduces ER stress induces by APAP overdose in in vivo models.** Proteomics analysis by LC-MS/MS was assessed in the liver of WT mice treated with a single dose of 360 mg/kg APAP and 24 h after APAP overdose mice were injected with siRNA *Cnnm4* ($n = 7$) or an unrelated control (siCtrl) ($n = 7$). **a** Volcano plot of specific proteins regulated by the absence of *Cnnm4* was shown. **b** GO process analysis for the regulated genes in APAP + siCtrl vs APAP + si*Cnnm4* labeled (APAP + siCtrl) and in APAP + si*Cnnm4* vs APAP + siCtrl labelled (APAP + si*Cnnm4*). A two-sided Student's-test was used to address the statistically significant differences between groups. No multiple testing correction procedure was applied. **c** mRNA levels of *Atf-6, Chop, X-box binding protein 1 (Xbp1s/Xbp1u) and binding immunoglobulin protein (Grp78)* in WT mice treated with a single dose of APAP 360 mg/kg and 24 h after siRNA *Cnnm4* ($n = 5$) or an unrelated control ($n = 5$). Values are represented as mean ± SEM. *$P < 0.05$; **$P < 0.01$; ***$P < 0.001$ (*Atf6* $P = 0.000482$ APAP vs Ctrl; $P = 0.00079$ APAP + si*Cnnm4* vs APAP) (*Chop* $P = 0.005$ APAP vs Ctrl; $P = 0.033$ APAP + si*Cnnm4* vs Ctrl) (*Xbp1* $P = 9.34$E-06 APAP vs Ctrl; $P = 0.00018$ APAP + si*Cnnm4* vs APAP) (*Grp78* $P = 0.00016$ APAP vs Ctrl; $P = 2.61$E-05 APAP + si*Cnnm4* vs APAP) (Student's test, two-sided). **d** Western blot analysis of phospho-EIF2α Ser 51 (p-EIF2α) and EIF2α ratio, β-ACTIN was used as a loading control. Quadrupled were used for experimental conditions. Values are represented as mean ± SEM. **$P < 0.01$; ***$P < 0.001$ ($P = 0.01$ APAP vs Ctrl; $P = 0.001$ APAP + si*Cnnm4* vs APAP) (Student's test, two-sided). **e** mRNA levels of *Atf-6, Chop, X-box binding protein 1 (Xbp1s/Xbp1u) and binding immunoglobulin protein (Grp78)* in WT mice treated with a single dose of APAP 360 mg/kg and 24 h after with a *Cnnm4* GalNAc siRNA molecule ($n = 4$) or an unrelated control ($n = 4$). Values are represented as mean ± SEM. *$P < 0.05$; **$P < 0.01$; ***$P < 0.001$ (*Atf6* $P = 0.01$ APAP + si*Cnnm4* vs APAP) (*Chop* $P = 0.02$ APAP vs Ctrl; $P = 0.001$ APAP + si*Cnnm4* vs Ctrl)

(*Xbp1* $P = 0.05$ APAP vs Ctrl; $P = 0.01$ APAP + si*Cnnm4* vs APAP) (*Grp78* $P = 0.0027$ APAP + si*Cnnm4* vs APAP) (Student's test, two-sided). **f** Western blot analysis of PERK, HSP90 was used as a loading control. Quadrupled were used for experimental conditions. Values are represented as mean ± SEM. **$P < 0.01$ ($P = 0.0025$ APAP vs Ctrl; $P = 0.0059$ APAP + si*Cnnm4* vs APAP) (Student's test, two-sided). **g** Cellular location of CNNM4 in WT mice liver tissue with an APAP overdose 360 mg/kg for 48 h and compared with a healthy mice. Protein expression was measured in membrane, mitochondria an ER location. CD36, voltage-dependent anion channel (VDAC), and Calnexin (CANX) were used as a loading controls. Triplicates were used for experimental conditions. Values are represented as mean ± SEM. **$P < 0.01$ (ER $P = 0.0059$ APAP vs Ctrl) (Student's test, two-sided). **h** Basal respiration using seahorse assay in mitochondria extract from mice liver with and 360 mg/kg of APAP overdose and 24 h later, injected with 3 mg/kg GalNAc Cnnm4 or an unrelated control. Values are represented as mean ± SEM. **$P < 0.01$ ($P = 0.0073$ APAP + GalNAc vs APAP) (Student's test, two-sided). Hepatocytes isolated from WT mice treated with a single dose of APAP 360 mg/kg and 24 h later, injected with 3 mg/kg GalNAc *Cnnm4* or an unrelated control. **i** The mitochondrial Reactive Oxygen Species was determined using MitoSOX staining. Values are represented as mean ± SEM. ***$P < 0.001$ ($P = 0.00038$ APAP + GalNAc vs APAP) (Student's test, two-sided) **j** ATP levels were measured. Values are represented as mean ± SEM. ***$P < 0.001$ ($P = 2.21$E-05 APAP + GalNAc vs APAP) (Student's test, two-sided) **k** ER tracker red staining. Quadrupled were used for experimental conditions. Values are represented as mean ± SEM. ***$P < 0.001$ ($P = 3.54$E-05 APAP + GalNAc vs APAP) (Student's test, two-sided). Source data are provided as a Source Data file.

## Competitive analysis for immunohistochemistry CNNM4 analysis

Paraffin-embedded liver and intestine samples were divided into four different groups, CNNM4 staining according to standard protocols described above, secondary antibody staining, and 1/1000 dilution CNNM4 antibody (Proteintech, 14066-1-AP) incubated with 0.3 μmol/mL of CNNM4$_{BATEMAN-cNMP-Ctail}$ and CNNM4$_{cNMP}$ pure proteins.

## Determination of ROS in liver tissue sections

Samples were sectioned in a cryostat (8 μm), and incubated with MnTBAP 150 μM at RT during 1 h. Samples were incubated with DHE (D-7008-10mg Sigma) 5 μM for 30 min at 37 °C. Section was mounted with mounting media containing DAPI.

## Isolation and culture of primary hepatocytes

Primary hepatocytes from WT mice were isolated by perfusion with collagenase type IV (Worthington). In brief, mice were anesthetized with isoflurane (1.5% isoflurane in $O_2$), the mouse's abdomen was opened and a catheter was inserted into the inferior vena cava and clamped supra inferior vena cava. Liver was perfused with buffer I (1× stock solution (D-Glucose, NaCl, KCl, and NaHCO$_3$), 5 mM EGTA) (37 °C, oxygenated), and portal vein was cut. Then, liver was washed with buffer II (1× stock solution) and subsequently, liver was perfused with buffer III (1× stock solution, 2 mM CaCl$_2$, collagenase type I (Worthington) (37 °C, oxygenated). After perfusion, the liver was placed in a petri dish containing 10% fetal bovine serum (FBS; Gibco) Minimum essential medium (MEM; Gibco) containing 1% penicillin (100 U/ml), streptomycin (100 U/ml), Amphotericin (100 U/ml) (Anti-Anti, Gibco) and glutamine (1%) (Gibco) and gently disaggregated with forceps. The liver was filtered through sterile gauze in a falcon. WT hepatocytes were washed twice in MEM medium (48 g, 5 min). Supernatant was discarded and the pellet was resuspended in fresh 10% FBS MEM supplemented with 1% PSA and 1% glutamine. Cell viability was validated by trypan blue considering 80% of viability for the experiments.

Isolated hepatocytes were seeded over collagen-coated 35 mm tissue culture dishes (5 × 10$^5$ cells/dish) in 10% FBS MEM supplemented with 1% PSA and 1% Glutamine and maintained in a 5% CO$_2$−95% air incubator at 37 °C during 6 h. After that, medium was replaced for fresh 0% FBS MEM supplemented with 1% PSA and 1% Glutamine overnight and different treatments were performed.

## Drug treatments in primary hepatocytes

APAP was dissolved in Dulbecco's phosphate-buffered saline (DPBS 10×, Gibco). Cells were treated with APAP at a dose of 10 mM for 1, 3, and 6 h.

## In vitro silencing

WT primary hepatocytes were transfected with 100 nM of C*nnm1* (Qiagen; siRNA Ref: S100954520), *Cnnm2* (Qiagen; siRNA Ref: S100954527), *Cnnm3* (Qiagen; siRNA Ref: S100954555) and *Cnnm4* siRNA (Sigma Aldrich; siRNA Ref: HA14051461) using Dharmafect (Dharmacon). Controls were transfected with an unrelated siRNA control (Qiagen). The second type of transfection was with 1 nM and 10 nM of *Cnnm4* siRNA (GalNAc#1 and GalNAc#2) (Silence Therapeutics). Hepatocytes were transfected for 12 h before the acetaminophen treatment.

## CNNM4 overexpression in primary hepatocytes

pDEST-26 vector mouse CNNM4 wild type and CNNM4 mutant Threonine 495 to Isoleucine cDNA kindly provided by Dr. Michel Tremblay (McGill University Goodman Cancer Research Centre, Montreal, Canada) were transfected into primary mouse hepatocytes using jetPRIME reagent (Polyplus) according to manufacturer's protocol. *Cnnm4* overexpression efficiency was evaluated by mRNA.

## Proteomics analysis by LC−MS/MS

Tissues were treated with cell lysis buffer (7 M urea, 2 M thiourea, 4% CHAPS), vortexed, and spined down to remove debris. Extracted protein was digested following the SP3 protocol described by ref. [61] with minor modifications. Trypsin was added to a trypsin:protein ratio of 1:10, and the mixture was incubated 2 h at 37 °C. Resulting peptides were dried out and resuspended in 0.1% formic acid. Samples were analyzed in a novel hybrid trapped ion mobility spectrometry-quadrupole time of flight mass spectrometer (timsTOF Pro with PASEF, Bruker Daltonics) coupled online to a EVOSEP ONE (EVOSEP). This mass spectrometer takes advantage of a novel

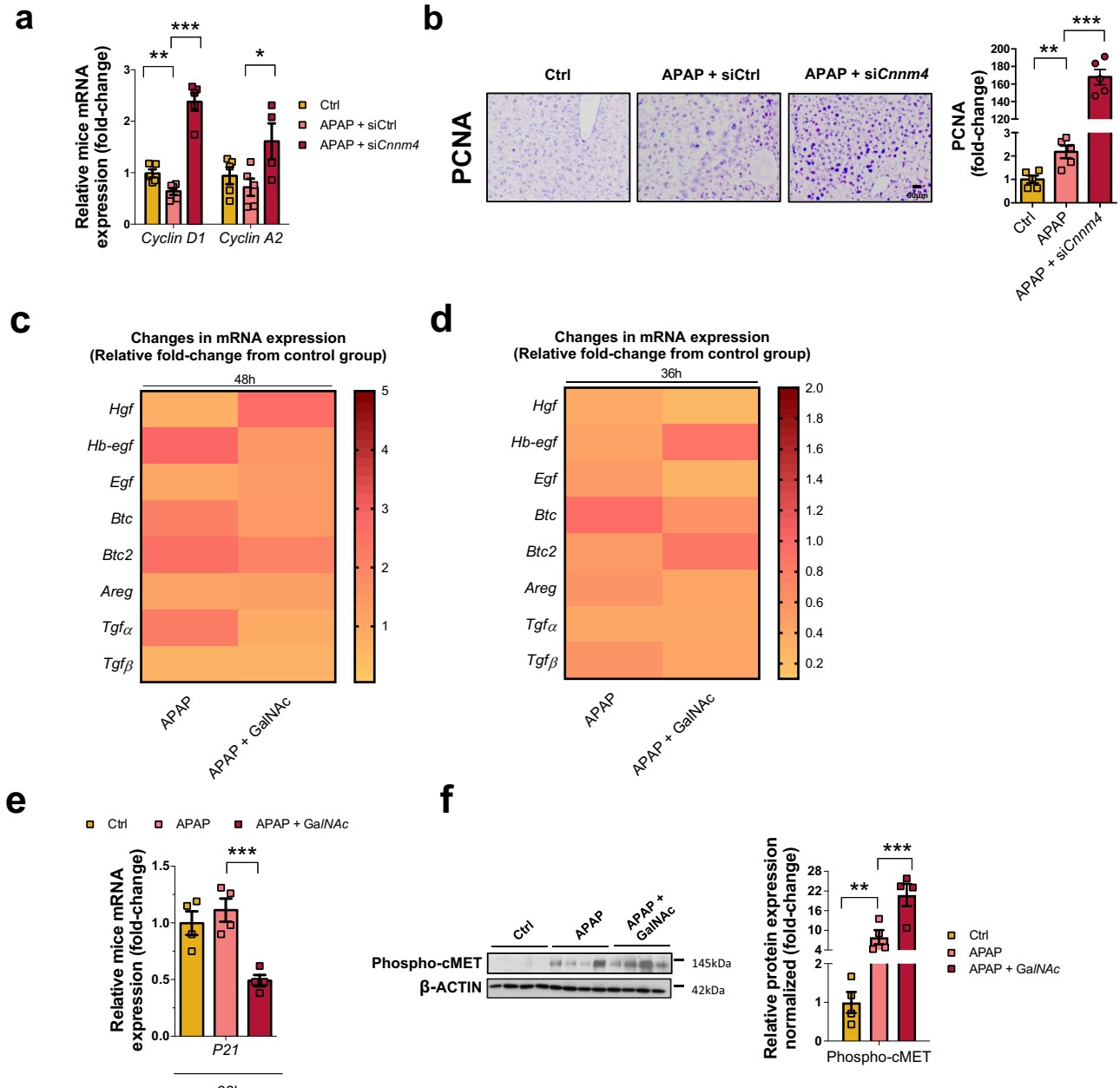

**Fig. 7 | *Cnnm4* knockdown induced liver regeneration in preclinical APAP model. a** mRNA levels of *Cyclin D1 and Cyclin D2*. Values are represented as mean ± SEM. **$P < 0.01$; ***$P < 0.001$ (Cyclin D1 $P = 0.01$ APAP vs Ctrl; $P = 1.31E\text{-}05$ APAP + si*Cnnm4* vs APAP) (Cyclin D2 $P = 0.042$ APAP + si*Cnnm4* vs APAP) (Student's test, two-sided). **b** PCNA expression by immunohistochemistry. Scale bar corresponds to 50 μm in WT mice treated for 48 h with APAP ($n = 5$) and silenced *Cnnm4* by siRNA the last 24 h of the hepatotoxic ($n = 5$). Changes in mRNA levels of Hepatocyte Growth Factor(*Hgf*), *Heparin Binding EGF Like Growth Factor (Hb-egf)*, Epidermal Growth Factor *(Egf)*, *Betacellulin (Btc)*, *Betacellulin v2 (Btc2)*, *Amphiregulin (Areg)*, *Transforming Growth Factor Alpha (Tgfα)*, and *Transforming Growth Factor Beta (Tgfβ)* in **c**. WT mice treated with a single dose of APAP 360 mg/kg for 48 h ($n = 5$) and treated with a si*Cnnm4* for the last 24 h ($n = 5$) and **d** WT mice treated

with a single dose of APAP 360 mg/kg for 36 h ($n = 4$) and treated with a *Cnnm4* GalNAc siRNA molecule the last 12 h ($n = 4$). **e** mRNA expression levels of p21 were determined in WT mice with an APAP overdose for 36 h ($n = 4$) and treated with *Cnnm4* GalNAc siRNA molecule for 12 h ($n = 4$). Values are represented as mean ± SEM. ***$P < 0.001$ ($P = 0.001$ APAP + GalNAc vs APAP) (Student's test, two-sided). **f** Western blot analysis of Phospho cMET (Tyr 1230/ Tyr1234/ Tyr1235) in WT mice with an APAP overdose for 36 h ($n = 4$) and treated with *Cnnm4* GalNAc siRNA molecule for 12 h ($n = 4$). β-ACTIN was used as a loading control. Values are represented as mean ± SEM. Values are represented as mean ± SEM. ***$P < 0.001$ ($P = 0.01$ APAP vs Ctrl; $P = 0.001$ APAP + GalNAc vs APAP) (Student's test, two-sided). Source data are provided as a Source Data file.

scan mode termed parallel accumulation-serial fragmentation (PASEF), which multiplies the sequencing speed without any loss in sensitivity[62]. Samples (200 ng) were directly loaded in a 15 cm analytical column (EVOSEP) and resolved at 300 nl/min with a 44 min gradient. Protein identification and quantification were carried out using PEAKS software using default settings. Searches were carried

out against a database consisting of mice protein entries (Uniprot/ Swissprot), with precursor and fragment tolerances of 20 ppm and 0.05 Da. Only proteins identified with at least two peptides at FDR < 1% were considered for further analysis. Data were loaded onto Perseus platform and further processed (log2 transformation, imputation)[63].

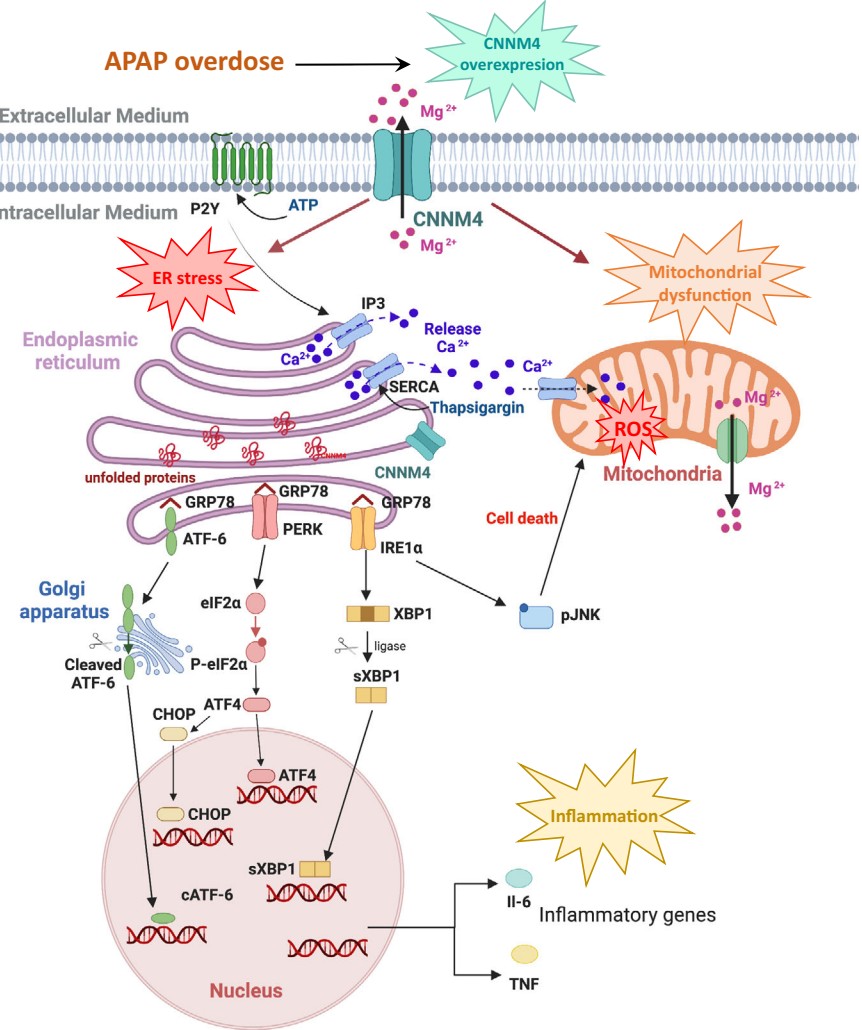

**Fig. 8 | Overall mechanisms involved in CNNM4 overexpression caused by APAP overdose.** In the liver, APAP overdose leads to increased expression of CNNM4 and an enrichment of its localization in the ER. Under these circumstances, reticulum stress and mitochondrial dysfunction are induced. The activity of CNNM4 is mainly influenced by its role as a magnesium effluxer. Therefore, increased CNNM4 expression reduces both mitochondrial and total cellular magnesium levels. When ER stress occurs, UPR markers, GRP78, ATF-6, PERK, pEIF2α, XBP1, and CHOP, are activated in order to solve the damage. Moreover, IRE1α pathway triggers pJNK activation in order to induce apoptosis. pJNK is translocated to mitochondria resulting in mitochondrial ROS, mitochondrial dysfunction, and consequently cell death. Another characteristic of ER stress is the $Ca^{2+}$ released into the cytoplasm to induce apoptosis. Finally, inflammation markers such as TNF and IL6 are induced as a consequence of APAP overdose. Silencing of *Cnnm4* in the liver was able to reduce mitochondrial dysfunction, ER stress and induce a regenerative response in the hepatocyte to overcome APAP overdose-induced necrosis. The image was created with BioRender.com.

## Protein isolation and Western blotting

Total protein extract from primary hepatocytes and hepatic tissue was resolved in sodium dodecyl sulfate-polyacrylamide gels and transferred to nitrocellulose membranes. Primary antibodies used are described in Supplementary Table III.

## RNA isolation and quantitative real-time polymerase chain reaction

Total RNA was isolated with Trizol (Invitrogen). 2 µg of total RNA was treated with DNAse (Invitrogen) and reverse transcribed into cDNA using MLV Reverse Transcriptase (Invitrogen). Quantitative real-time PCR (RT-PCR) was performed using SYBR Selected Master Mix (Applied Biosystems) and Viia7 Real-Time PCR System (Applied Biosystems). The Ct values were extrapolated to a standard curve, and the data were then normalized to the housekeeping expression (*Arp*). Primers (Sigma Aldrich) are described in Supplementary Data 1.

## TUNEL assay

TUNEL assay was performed in primary hepatocytes using the in situ cell death detection kit (Roche) according to the manufacturer's instructions and described in ref. 12. Five images were taken for experimental condition. Quantification of cell death vs total nucleus was performed using FIJI (ImageJ).

## RealTime-Glo Annexin V apoptosis and necrosis assay

WT mice primary hepatocytes and Thle2 cell line hepatocytes were transfected as described in the section "*In vitro silencing*" and treated with 10 mM of APAP for 3 h. Apoptosis and necrosis processes were evaluated by Annexin V Apoptosis and Necrosis Assay (Promega Corporation, Madison, WI) according to the manufacturer's instructions.

## CYP2E1 activity

Primary hepatocytes were grown in 12-well plates. After with 10 mM of APAP for 3 h, growth medium was removed and cells were washed with

saline Hepes buffer previously warmed at 37 °C. Then, cells were incubated for an hour with a solution of 50 μM Chlorzoxazone (Sigma Aldrich, C4397) in Saline Hepes Buffer. After the incubation, containing medium was collected in new tubes and analyzed by LC-MS/MS.

## Subcellular extraction of proteins from fragmented tissue

Healthy mice and mice were treated with APAP in a single dose of 360 mg/kg, and 48 h later, mice were sacrificed and mice liver tissue was fragmented. Subcellular fractionation was performed by ProteExtract Subcellular Proteome Extraction Kit (Millipore) according to manufacturer's protocol.

## Endoplasmic reticulum and mitochondria isolation

Isolation of ER and mitochondria was performed in 1 g of liver tissue of control animals and those treated with APAP 360 mg/kg for 48 h according to manufacturer's protocol described in endoplasmic reticulum isolation kit (Sigma Aldrich, ER0100).

## Determination of the mitochondrial membrane potential

Mitochondrial membrane potential was determined in WT primary hepatocytes after 10 mM of APAP for 3 h and incubated with 0.5 μM of Tetramethyldhodamine Ethyl Ester Perchlorate (TMRE, ThermoFisher Scientific) for 30 min at 37 C in a $CO_2$ incubator. Fluorescence was read at 548 nm (excitation) and 574 nm (emission) using a plate reader Spectra M2 (BioNova).

## Determination of mitochondrial Reactive Oxygen Species (ROS)

Mitochondrial ROS production in primary hepatocytes was determined by MitoSOX Red mitochondrial superoxide indicator (Invitrogen, USA). Fluorescence was read at 510 nm (excitation) and 595 (emission) using a plate reader Spectra M2 (BioNova).

## Intracellular magnesium and cytosolic calcium determination

Primary hepatocytes grown in glass coverslips in 24-well plates, were loaded with 2 μM Mag-S-AM[41] and Mg-S-TTP-AM for intracellular $Mg^{2+}$ determination and FURA-2 for cytosolic calcium determination, in 0% FBS MEM and incubated at 37 °C and 5% $CO_2$ during 30 min. Then, the containing medium was removed and fresh 0% FBS MEM was added and incubated for 1 h. Then, Coverslips were washed in a 20 mM Tris-HCl, 2.4 mM $CaCl_2$, 10 mM glucose, pH 7.4 buffer and mounted on a thermostatized perfusion chamber on an Eclipse TE 300-based microspectrofluorometer (Nikon) and visualized with a 40× oil-inmersion fluorescence. Intracellular $Mg^{2+}$ was determined by comparing the relative fluorescence ratio between the $Mg^{2+}$-labelled probe, light excited at 340 nm, and the not-labelled probe, excited at 380 nm. The excited light ratio was determined with a Delta system (Photon Technologies International). Intracellular $Ca^{2+}$ levels were determined using the method described by ref. [43].

## Endoplasmic reticulum (ER)-tracker red staining

ER-tracker Red (ThermoFischer) assay was performed in primary hepatocytes according manufacturer's instructions. Covers were mounted with Fluoroshield Mounting Medium with DAPI (Dako). Pictures were taken with an Axio Imager D1 Upright Fluorescence Microscope (Carl Zeiss AG, Jena). Quantification of fluorescence intensity was performed using FIJI (ImageJ). Five images were taken per experimental condition. Five images were taken for experimental condition.

## Respiration studies in primary hepatocytes and liver mitochondria

The respiration of primary hepatocytes was measured at 37 °C by high-resolution respirometry with the Seahorse Bioscience XF24-3 Extracellular Flux Analyzer. For the measurement of the oxygen consumption rate (OCR), as the rate change of dissolved $O_2$, primary mouse hepatocytes were seeded respectively in a collagen I coated XF24 cell

culture microplate (Seahorse Bioscience), at $2.0 \times 10^4$ cells per well. WT hepatocytes were transfected in with 100 nM of Cnnm4 siRNA (Qiagen) using Dharmafect (Dharmacon). Controls were transfected with an unrelated siRNA control (Qiagen). The day after, growth medium was removed and replaced with 500 μl of assay medium prewarmed to 37 °C, composed of MEM without bicarbonate containing 1 mM sodium pyruvate, 2 mM L-glutamine, and maintained at 37 °C in a chamber without $CO_2$. After 1 h, some experimental groups were stimulated with 10 mM of APAP overdose. Afterward, Oxygen Consumption Rate (OCR) changes associated with APAP stimulation were assessed during 3 h and then, sequential injections were performed through ports in the XF Assay cartridges. The following pharmacologic inhibitors were used: Oligomycin (1 mM), an inhibitor of ATP synthase, which allows the measurement of ATP-coupled oxygen consumption through OXPHOS; carbonyl cyanide 4-trifluoromethoxy-phenylhydrazone (FCCP) (300 nM), an uncoupling agent that allows maximum electron transport, and therefore a measurement of the maximal OXPHOS respiration capacity; and Rotenone (1 μM), a mitochondrial complex I inhibitor. Upon the sequential delivery of the inhibitors, changes in OCR were recorded. At the end of the experiments, OCR measurements were normalized to cell number by using crystal violet staining.

## Trypan blue

Trypan blue assay was performed in primary hepatocytes using 0.2% of the Trypan Blue solution (Sigma-Aldrich) according to the manufacturer's instructions. Five images were taken for experimental condition.

## Treatments in THLE2 cell line

The human liver epithelial cells from ATCC® (CRL-2706) were cultured with BEGM medium from Lonza/Clonetics Corportation (BEGM Bullet Kit; CC3170), discarding gentamycin/Amphotericin (GA) and Epinephrine. 5 ng/mL EGF, 70 ng/mL Phosphoethanolamine, 10% fetal bovine serum, 1% PSA and 1% Glutamine. THLE2 hepatocytes were transfected by overnight incubation with two different siRNA sequences: 100 nM of Cnnm4 siRNA#1 (Life technologies; siRNA Ref: s25465) and 100 nM of Cnnm4 siRNA#2 (Integrated DNA Technologies, Inc; Ref: 229567381) using Dharmafect (Dharmacon). Controls were transfected with an unrelated siRNA control (Qiagen). Cells were treated with APAP at a dose of 10 mM for 3 and 6 h.

## ELISA TNF and IL6

To analyze the effect of TNF and IL-6, serum mice were analyzed by ELISA assay (BD Biosciences, Mouse TNF ELISA Set II and Invitrogen, Mouse IL-6 ELISA Kit, respectively) according to the manufacturer's instructions.

## Quantification of GSSG and GSH levels

Liver samples were analyzed with a UPLC system (Acquity, Waters, Manchester) coupled to a Time of Flight mass spectrometer (ToF MS, SYNAPT G2, Waters). A $2.1 \times 100$ mm, 1.7 mm BEH amide column (Waters), stabilized at 40 °C, was used to separate the analytes before entering the MS. The aqueous phase consisted of 99.5% water, 0.5% formic acid, and 20 mM ammonium formate while the organic phase consisted of 29.5% water, 70% MeCN, 0.5% formic acid, and 1 mM ammonium formate. The extracted ion trace was obtained for GSH ($m/z = 308.0916$) and GSSG ($m/z = 613.1598$) in a 20 mDa window and subsequently smoothed and integrated with QuanLynx software (Waters, Manchester).

## Expression and purification of recombinant CNNM4$_{BATEMAN-cNMP-Ctail.(356-775)}$

The vector with the construct was transformed into Escherichia coli BL21 (DE3) competent cells (Thermofisher), through thermal shock technique, mixing 10 ng of vector with 50 μL of competent cells. The

mixture was kept in ice for 30 min. Subsequently, the cells received a thermal shock at 42 °C for 45 s. Then, cells were transferred to ice for 2 min and after that, 500 μl of Luria-Bertani (LB) medium (previously attemperated at 37 °C) was added and incubated at 37 °C for 1 h in constant agitation in an incubator Excella E24R (New Brunswick Scientific). Subsequently, cells were centrifuged at 18,000 g during 1 min in a centrifuge Microfuge 22 R (Beckman Coulter) at 4 °C. The supernatant was discarded and cells were seeded in a petri dish with LB Agar and 100 μg mL$^{-1}$ of ampicillin at 37 °C overnight. Next day, a bacteria colony was grown in 150 mL of LB medium with 100 μg mL$^{-1}$ of ampicillin at 37 °C overnight in constant agitation. This culture was used to grow the culture at large-scale. Bacterial cells were grown (37 °C, 200 rev min$^{-1}$) in six 5 L flasks containing 2 L of autoclaved LB and supplemented with 100 μg mL$^{-1}$ of ampicillin and 20 mL of cell culture grown overnight. Once OD$_{600}$ reached 0.8, expression was induced with 0.5 mM IPTG at 20 °C O/N. Next day, the bacterial pellet was centrifuged at 5053 g for 20 min. The cell pellet was resuspended in a lysis buffer (50 mM Tris-HCl pH 8, 200 mM NaCl, 50 mM Imidazole, 0.1 mM phenylmethylsulfonium fluoride (PMSF), 1 mM Benzamidine, 1 mM tris (2-carboxyethyl)phosphine (TCEP) and DNAse) and sonicated (10 cycles of 15 s with 60% of amplitude) in a sonicator Sonics Vibracell VCX500. Cell lysate was harvested by subsequent centrifugation at 186,009 g for 1 h at 4 °C in a centrifuge Avanti J-26 XP (Beckman Coulter) and 45TI rotor (Beckman Coulter). The supernatant, previously filtered in a filter of 0.45 μm, was loaded on a 5 mL HiTrap HP column (GE healthcare) previously equilibrated in a buffer with (50 mM Tris-HCl pH 8, 200 mM NaCl, 50 mM Imidazole, 1 mM TCEP). The bound protein was subsequently washed with at least five column volumes (CV) of wash buffer (50 mM Tris-HCl pH 8, 200 mM NaCl, 1 mM TCEP) and then eluted with 300 mM imidazole in the wash buffer. The presence of CNNM4 $_{BATEMAN-cNMP-Ctail}$ was confirmed by electrophoresis technique with SDS-PAGE acrylamide gel. The purest fractions were mixed with 1 mg of TEV (Tobacco Etch Virus) per 30 mg of target protein and were dialysated in the exchanged buffer (50 mM Tris pH 8, 150 mM NaCl, 1 mM TCEP) at 4 °C overnight in constant agitation. Then, the supernatant was loaded on a 5 mL HiTrap HP column (GE healthcare) previously equilibrated with buffer (50 mM Tris-HCl pH 8, 200 mM NaCl, 50 mM Imidazole, 1 mM TCEP). This second chromatographic step had the objective to discard the 6xHis tag of the target protein and TEV protease, which also contains a 6xHis tag. The flow-through fractions were collected and confirmed by SDS-PAGE acrylamide gels. Subsequently, the sample was concentrated to 2 mL approximately by centrifugation at 3901 g at 4 °C using a concentrator Amicon Ultra-15 10K. This solution was loaded in a Hi Load 16/60 Superdex-75 column (Ge Healthcare) previously equilibrated in a buffer (50 mM HEPES pH 7.4, 150 mM NaCl, 1 mM TCEP). The protein elution was realized at 0.3 mL min$^{-1}$ in 1CV. The presence of CNNM4 $_{BATEMAN-cNMP-Ctail}$ was confirm by SDS-PAGE (Supplementary Fig. 9) and mass spectrometry. The purest fractions of the target protein were selected and concentrated in an Amicon Ultra-15 10K concentrator to ~26 mg mL$^{-1}$. The protein was aliquoted in 26 μL, frozen by liquid nitrogen, and stored at −80 °C before processing. The fractions of the purified protein are represented in a gel (Supplementary Fig. 1A).

### Expression and purification of recombinant CNNM4 $_{cNMP\ (545-730)}$

The mus musculus CNNM4$_{cNMP}$ construct is composed by the residues 545–730 of the cNMP domain. The vector with the construct was transformed into Escherichia coli BL21 Star (DE3) competent cells (Thermofisher), through thermal shock technique following the same protocol as CNNM4$_{BATEMAN-cNMP-Ctail}$.

The expression and purification of CNNM4$_{cNMP}$ followed the same protocol as CNNM4$_{BATEMAN-cNMP-Ctail}$, with the difference that bacteria cells were grown at 37 °C with 1 mM IPTG induction during 4 h. The protein was aliquoted, snap-frozen in liquid nitrogen and stored at

−80 °C before processing. The fractions of the purified protein are represented in a gel (Supplementary Fig. 1B).

### Statistical analysis

Statistical significance was performed using GraphPad Prism 6 software. Data were represented ±SEM for each experimental group ($n > 3$). Stadistical significance was determined by Student's $t$ test (when two groups were compared) or one-way analysis of variance ANOVA (when more than two groups were compared) followed by post hoc Bonferroni test. A $p < 0.05$ was considered as stadistically significant.

### Reporting summary

Further information on research design is available in the Nature Research Reporting Summary linked to this article.

## Data availability

The mass spectromic data generated in this study have been deposited in the PRIDE database under accession code PXD036682.

All other data generated during and/or analyzed during the current study are included in this published article (and its supplementary information files), available from the corresponding author on reasonable request. Source data are provided with this paper.

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

## Acknowledgements

This work was supported by Ministerio de Ciencia, Innovación y Universidades MICINN: PID2020-117116RB-I00 integrado en el Plan Estatal de Investigación Científica y Técnica y Innovación, cofinanciado con Fondos FEDER (to MLM-C), Ministerio de Ciencia e Innovación CONSOLIDER-INGENIO 2010 Program Grant CSD2008-00005 (to LAM-C); Spanish Ministry of Economy and Competitiveness Grant BFU2013-47531-R, BFU2016-77408-R, PID2019-109055RB-100 (to L.A.M.-C.) (MINECO/FEDER, UE); Asociación Española contra el Cáncer (MLM-C, TC-D), Fundación Científica de la Asociación Española Contra el Cáncer (AECC Scientific Foundation) Rare Tumor Calls 2017 (to M.L.M.-C.), La Caixa Foundation Program (to M.L.M.-C.), Fundacion BBVA UMBRELLA project (to M.L.M.-C.), Ayuda RYC2020-029316-I financiada por MICIN/AEI/10.13039/501100011033 (to TC-D), Plataforma de Investigación Clínica-SCReN (PT17 0017 0020) (to M.I.-L.), programa retos RTC2019-007125-1 (to M.L.M.-C, J.S.), Proyectos Investigacion en Salud DTS20/00138 (to M.L.M.-C., J.S) ERA-Net E-Rare EJP RD Joint Translational Call for Rare Diseases FIGHT-CNNM2 (EJPRD19-040) and from Instituto Carlos III, Spain (REF G95229142) (to L.A.M.-C.), US National Institutes of Health under grant CA217817 (to D.B.), Ciberehd_ISCIII_MINECO is funded by the Instituto de Salud Carlos III. We thank MINECO for the Severo Ochoa Excellence Accreditation to CIC bioGUNE (SEV-2016-0644) and PhD fellowship from MINECO (REF BES-2017-080435) awarded to I.G.-R. The collection and storage of patients tissues was supported by the Newcastle Biomedicine Biobank and the European Community's Seventh Framework Programme (FP7/2001–2013) and Cancer Research UK awards Cancer Research UK grants C18342/A23390; C9380/A18084 and C9380/A26813. Finally, we would like to acknowledge Begoña Rodríguez Iruretagoyena for the technical support provided.

## Author contributions

Conceptualization: I.G.-R., L.A.M.-C., M.L.M.-C., D.B. Funding acquisition: T.C.-D., L.A.M.-C., M.L.M.-C. Experiments: I.G.-R., J.S., N.G.-U., M.S.-M., M.M.-G., R.R.-A., S.L.-O., C.G.-P., C.F.-R., D.-C., M.U.-L., L.A., J.A., T.C.-D., P.I., J.C., S.H., P.D.-P., R.J., M.A.-A., C.M., U.S., M.L.-T., J.W.-D., S.-M., M.-V.M., H.-R., R.J.-A., M.I.-L. Supervision: L.A.M.-C., M.L.M.-C., D.B. Writing—Original draft, review and editing: I.G.-R., L.A.M.-C., M.L.M.-C., D.B. All authors have revised and approved the final version of the manuscript.

## Competing interests

The rest of the authors declare no competing interest. U.S. is employed by Silence Therapeutics GmbH. This study is under the patent "Nucleic acids for inhibiting expression of Cnnm4 in a cell". WO2021239825A1. Inventor: Ute Schaeper, Sibylle Dames, Steffen Schubert, Alfonso Martínez de la Cruz, Jorge Simón Espinosa, Irene González Recio, María Luz Martínez Chantar. (02/12/2021). The present patent previously described identifies the use of Galnac siRNA Cnnm4 for the treatment of liver disease. This article reinforces the validation of this therapeutic approach for the pathology of DILI due to paracetamol overdose.

## Additional information

Irene González-Recio[1], Jorge Simón[1,2], Naroa Goikoetxea-Usandizaga[1], Marina Serrano-Maciá[1], Maria Mercado-Gómez[1], Rubén Rodríguez-Agudo[1], Sofía Lachiondo-Ortega[1], Clàudia Gil-Pitarch[1], Carmen Fernández-Rodríguez[1], Donatello Castellana[3], Maria U. Latasa[4], Leticia Abecia[5,6], Juan Anguita[5,7], Teresa C. Delgado[1], Paula Iruzubieta[8], Javier Crespo[8], Serge Hardy[9,10], Petar D. Petrov[2,11], Ramiro Jover[2,11], Matías A. Avila[2,4], César Martín[12], Ute Schaeper[13], Michel L. Tremblay[9,10], James W. Dear[14], Steven Masson[15,16],

**Misti Vanette McCain** [15], **Helen L. Reeves** [15,16], **Raul J. Andrade** [2,17], **M. Isabel Lucena** [2,18], **Daniela Buccella** [19] ✉, **Luis Alfonso Martínez-Cruz** [1] ✉ **& Maria L Martínez-Chantar** [1,2] ✉

[1]Liver Disease Lab, Center for Cooperative Research in Biosciences (CIC bioGUNE), Basque Research and Technology Alliance (BRTA), Bizkaia Technology Park, Building 801A, 48160 Derio, Spain. [2]Centro de Investigación Biomédica en Red de Enfermedades Hepáticas y Digestivas (CIBERehd), Carlos III National Health Institute, Madrid, Spain. [3]Research & Development, Center for Cooperative Research in Biosciences (CIC bioGUNE), Basque Research and Technology Alliance (BRTA), Bizkaia Technology Park, Building 801A, 48160 Derio, Spain. [4]Hepatology Programme, CIMA, Idisna, Universidad de Navarra, Avda, Pio XII, n 55, 31008 Pamplona, Spain. [5]Inflammation and Macrophage Plasticity Laboratory, Center for Cooperative Research in Biosciences (CIC bioGUNE), Basque Research and Technology Alliance (BRTA), 48160 Derio, Bizkaia, Spain. [6]Departamento de Inmunología, Microbiología y Parasitología, Facultad de Medicina y Enfermería. Universidad del País Vasco/ Euskal Herriko Unibertsitatea (UPV/EHU), Barrio Sarriena s/n 48940, Leioa, Spain. [7]IKERBASQUE, Basque Foundation for Science, Bilbao, Spain. [8]Gastroenterology and Hepatology Department, Marqués de Valdecilla University Hospital, Clinical and Translational Digestive Research Group, IDIVAL, Santander, Spain. [9]Department of Biochemistry, McGill University, H3G 1Y6 Montréal, QC, Canada. [10]Rosalind and Morris Goodman Cancer Research Centre, McGill Unversity, H3A 1A3 Montréal, QC, Canada. [11]Experimental Hepatology Joint Research Unit, IIS Hospital La Fe & Dep. Biochemistry, University of Valencia, Valencia, Spain. [12]Biofisika Institute (UPV/EHU, CSIC) and Department of Biochemistry and Molecular Biology, University of the Basque Country (UPV/EHU), 48940 Leioa, Spain. [13]Silence Therapeutics GmbH, Berlin, Robert Rössle Strasse 10, 13125 Berlin, Germany. [14]Pharmacology, Toxicology and Therapeutics, Centre for Cardiovascular Science, University of Edinburgh, Edinburgh, UK. [15]The Liver Unit, Newcastle-upon-Tyne Hospitals NHS Foundation Trust, Newcastle upon Tyne NE7 7DN, UK. [16]Newcastle University Translational and Clinical Research Institute, The Medical School, Newcastle University, Newcastle upon Tyne NE2 4HH, UK. [17]Unidad de Gestión Clínica de Enfermedades Digestivas, Instituto de Investigación Biomédica de Málaga-IBIMA, Hospital Universitario Virgen de la Victoria, Universidad de Málaga, Málaga, Spain. [18]Servicio de Farmacología Clínica, Instituto de Investigación Biomédica de Málaga-IBIMA, Hospital Universitario Virgen de la Victoria, UICEC SCReN, Universidad de Málaga, Málaga, Spain. [19]Department of Chemistry, New York University, New York, NY 10003, USA. ✉e-mail: dbuccella@nyu.edu; amartinez@cicbiogune.es; mlmartinez@cicbiogune.es

