## [Peer Review File · Nature Communications]

Title: Restoring cellular magnesium balance through Cyclin M4 protects against acetaminophen-induced liver damageREVIEWER COMMENTS

Reviewer #1 (Remarks to the Author):

There are several issues:

Human studies: There is very little description of the patients and controls. There should at least be a RUCAM score to indicate that the idiosyncratic cases were likely DILI cases. The average ALT for the patients with hepatocellular idiosyncratic DILI was only 120 ± 64 . That is very mild liver injury. However, it appears that the only data from the idiosyncratic DILI patients is serum Mg, which makes it less of an important issue. It is important that the authors always make it clear when the data are from idiosyncratic DILI patients vs acetaminophen overdose patients. Why was a liver biopsy performed on patients with an acetaminophen overdose? How would it change therapy? How long after the overdose was the biopsy performed? It is stated that "Biological samples from four controls with no liver diseases and tolerant to commonly used drugs attending routine work monitoring were used." In figure 1H there is liver immunohistochemical staining of CNNM4 in 13 acetaminophen overdose patients compared to a healthy group. Presumably the liver sample from the healthy group was obtained by a liver biopsy; I don't know how else material for liver immunohistochemistry could be obtained from the healthy group. A liver biopsy is associated with significant serious risks. It would be very unethical to perform a liver biopsy on a healthy person, and it is hard to believe that it would be part of "routine work monitoring". What type of work? Is there something I am missing?

The authors seem to equate idiosyncratic DILI with acetaminophen-induced DILI. Acetaminophen DILI is a direct toxin that causes acute toxicity while idiosyncratic DILI is idiosyncratic, delayed in onset, and in most cases mediated by the adaptive immune system, specifically CD8 T cells. The implication is that because CNNM4 decreases acetaminophen DILI in animals it would also work for the treatment of idiosyncratic DILI. It seems very unlikely that inhibition of CNNM4 would prevent liver injury mediated by CD8 T cells involving perforin and granzymes. I could be wrong, but that hypothesis is not tested in this paper, and it would be difficult to test clinically. Just because serum Mg increases does not mean that CNNM4 is involved; it could just be the result of cell death. Without additional evidence for involvement of CNNM4 in the mechanism of idiosyncratic DILI, I think it is inappropriate to imply that inhibition of CNNM4 would be an effective treatment of this condition.

In my opinion, the most important data is the protection by siRNA Cnm4. If I understand correctly, the mice were treated with acetaminophen and 24 hours later they were treated with siRNA, but the of siRNA is not clear. If I understand correctly, the ALT and histology in Figure 5 are 48 hours after the acetaminophen administration and 24 hours after the siRNA administration. I would have liked to see a comparison between the effects of siRNA and glutathione on toxicity since glutathione is the standard therapy for acetaminophen overdoses. I would not expect that glutathione would be very effective 24 hours after the acetaminophen, but I am not sure.

The timing in the in vitro experiments is not entirely clear; was the siRNA added at the same time as the acetaminophen? Although I am not an expert in many of the pathways studied in vitro, the data appears

to support the basic hypothesis in the case of acetaminophen, and I think that does represent an important contribution.

In summary, it seems counter intuitive that a toxin would upregulate a transporter that would increase the toxicity of that toxin, but the authors provide a large amount of data to support that conclusion. I do not believe that the results of studies with acetaminophen can be extrapolated to idiosyncratic DILI. I am very worried about the ethics of the human studies, especially the source of liver samples from normal controls. The native language of the authors is clearly not English, and that makes it more difficult to read; the manuscript requires some editing.

Jack Uetrecht

Reviewer #2 (Remarks to the Author):

This study by Gonzales-Recio et al reports that CNNM4 expression is elevated in the liver of acetaminophen-induced liver injury patients and its model mice. CNNM4 is a Mg²⁺-extruding transporter and its overexpression perturbed mitochondrial Mg²⁺ homeostasis, resulting in augmentation of ROS levels and ER stress. Furthermore, suppression of CNNM4 expression by in vivo delivery of CNNM4 siRNA protected liver from injury by restoring Mg²⁺ homeostasis.

Identification of CNNM4 as a potential therapeutic target for liver injury is intriguing. However, I have several serious concerns regarding the novelty of this study and the experimental validation of the findings.

Major issues:

The same group has reported hepatic CNNM4 overexpression in NASH patients and its model mice (J. Hepatol. 2021, PMID: 33571553). The main conclusion of the NASH paper is very similar to that of this manuscript, but it is not described in this manuscript. The authors should explain the details by citing the previous paper in introduction and discussion, and indicate novel points revealed in this study. In addition, the same paper is actually cited twice as refs. 30 and 37 in different contexts, which should also be corrected appropriately.

The conclusion heavily depends on RNAi experiments in cultured cells and mice. In general, two or more siRNAs targeting different sequences are needed for validation. Moreover, definitive conclusions cannot be drawn solely from such type of experiment, which is often associated with non-specific effects. The authors should also take some different approaches (e.g. gene knockout), to firmly establish the importance of CNNM4 in liver injury.

It is difficult to quantitate CNNM4 expression levels by immunohistochemistry. In addition, how did the

authors verify the signals? CNNM4 is reportedly localized at the basolateral membrane of the epithelial cells in the intestine (PLOS Genet. 2013, PMID: 24339795), but the staining pattern shown in Supplemental Figure 4B does not seem to coincide with that. Immunofluorescence staining of intestinal sections containing the crypt structure, which are obtained from control and siCnm4-treated mice (without APAP treatment), will be the best to confirm the authenticity of the signals. Also, the authors should perform immunoblotting experiments for verification and precise quantitation.

It was previously reported that CNNM4 knockdown results in mitochondrial ROS generation in mammalian cultured cells (Antioxid. Redox Signal. 2020, PMID: 32148064), which seems to be contradictory to the conclusion of this manuscript. The authors should discuss about the discrepancy.

Figure 2D, if Mg^{2+} decrease caused by CNNM4 overexpression induces hepatocyte damage, Mg^{2+} supplementation at appropriate levels should restore the damage. Is the addition of 5 mM Mg^{2+} in the medium sufficient? Did the authors check the cytosolic and mitochondrial Mg^{2+} levels? The authors should also test the effect of artificial overexpression of CNNM4 on hepatocyte damage, such as cell death, ROS augmentation, and ER stress. CNNM4 mutants lacking the Mg^{2+} transport activity would be a good negative control.

Figure 3A, Mg^{2+} levels are strongly decreased in the mitochondria. This is very curious because CNNM4 proteins are supposedly localized at the plasma membrane. Is there any mechanism to maintain cytosolic Mg^{2+} levels at the expense of mitochondrial Mg^{2+} ? Alternatively, CNNM4 proteins may be partly localized at the mitochondrial membrane and regulate Mg^{2+} homeostasis in the mitochondria, as illustrated in Figure 3. The authors should address this intriguing problem by experimentation.

Minor issues:

Figure 4A, the data are not clear with large variations and seem to be heavily modified by image-processing software.

Figure 4C, what is the meaning of S6 phosphorylation?

Figure 4D, some explanation is needed for Ca^{2+} oscillation by thapsigargin treatment, by citing appropriate papers.

Supplemental Figure 1C, lines connecting each data point should be removed.

Supplemental Figure 4B, CNNM4 expression levels are greatly reduced by APAP treatment, but why? Also, the non-effectivity of siCnm4 may be due to this decrease.

Supplemental Figure 4C, is there any statistical difference between "APAP + siCtrl" and "APAP + siCnm4"?

Scale bars in microscopic images are too small in many figures.

Reviewer #3 (Remarks to the Author):

Authors investigated the role of transporters family Cyclin M4, also known as CNNM4 that is a key role in the Mg²⁺ transport across the cell membranes, in acetaminophen (APAP)-induced liver injury. They showed increased expression of CNNM4 in APAP-treated cultured hepatocytes, mouse livers and liver samples of patients with APAP-induced liver injury (AILI), and these patients also had increased serum Mg²⁺ levels. More importantly, silencing CNNM4 in the liver with nanoparticle- and GalNAc-formulated siRNA protected against liver from AILI. While to study Mg²⁺ in APAP is novel and the study has clinical and translation value, there were major concerns over the lack of rigor and robustness of this study. The mechanistic parts were also weak and not sure whether ER stress would be the primary mechanism to explain the protective effects of CNNM4 knockdown.

The in vitro studies were performed after 1-6 hrs treatment of APAP (which would be considered as early event), and in vivo APAP model was 48hrs post treatment (considered as late event). It is unclear how the mechanisms studied in vitro relevant to the in vivo mouse studies? If ER stress is key mechanism downstream of Cnnm4 knockdown, then targeting ER stress per se should also have beneficial effects for this APAP injury mouse model?

Major concerns:

1. There were concerns over the data rigor and robustness. There was lacking of experimental details in the figure legend, and many animal experimental data were only from 3 mice. For instance, for many experiments, the dose of APAP and whether the mice were fasted or not were not described.
2. Figure 1, one the most key piece of data presented in Figure 1A were from 10 mixed DILI patients. While this may tend to suggest that increased serum Mg²⁺ may be a general event, it will be nice to show how specific this to APAP-induced live injury. Authors may consider to obtaining more samples just from APAP liver injury patients to specifically address its correlation with APAP.
3. Figure 1, in addition to mRNA and IHC staining, the increased CNNM4 should be confirmed by western blot analysis (which will be more quantitative at the protein levels). In addition, it will be more insightful if they authors can show where are the cellular locations of CNNM4 after APAP treatment.
4. Figure 2, what were the concentrations of APAP used in these cell cultures studies? How the TUNEL positive cells were quantified? More robust and objective assays such as LDH leakage should be considered. More importantly, the knockdown and overexpression efficiency of Cnnm4 should be validated. For all in vitro studies, the APAP metabolism and bioactivation after Cnnm4 knockdown should be determined as this is critical to rule out its effects on APAP metabolism.
5. Figure 4, in addition to the mRNA changes of the UPR, some of them should be confirmed at the protein levels.
6. Figure 5, the si Cnnm4 alone group was missing. More importantly, the knockdown efficiency of Cnnm4 in the mouse liver should be validated by western blot analysis. More detail mechanisms were missing in this import part of in vivo work on how knockdown of Cnnm4 protected against APAP liver injury. Previously showed mitochondrial damage should be examined in the in vivo study to highlight the

relevance of their in vitro findings. As this model is post APAP 48 hrs, it is known at this time point the mouse has passed the injury phase and most likely enter the recovery phase. The impact of cnmm4 knockdown on increased liver regeneration (PCNA staining) is interesting but how knockdown of cnmm4 increase cell proliferation was not determined in detail.

7. Figure 6, the caspase 3 activity and apoptosis data interpretation was questionable as it is well known APAP induced necrosis and caspase activation is not relevant in this model.

8. It was unclear why the manganese superoxide dismutase activities after manipulating cnmm4 was not determined? Which may be an important protective mechanism against APAP liver injury of cnmm4 knockdown. It was also unclear how these inflammatory gene expression changes would be mechanistically linked to hepatocytes proliferation and regeneration.

Reviewer #4 (Remarks to the Author):

My expertise is limited to the siRNA therapeutics and not liver injury. I wish to comment solely on the design of the RNA molecule.

It is not immediately obvious if sequence design and stability was considered. These are likely downstream considerations of the present study, but are certainly worth a short discussion.

REVIEWER COMMENTS

Reviewer #1 (Remarks to the Author):

There are several issues:

Human studies: There is very little description of the patients and controls. There should at least be a RUCAM score to indicate that the idiosyncratic cases were likely DILI cases. The average ALT for the patients with hepatocellular idiosyncratic DILI was only 120 ± 64 . That is very mild liver injury. However, it appears that the only data from the idiosyncratic DILI patients is serum Mg, which makes it less of an important issue. It is important that the authors always make it clear when the data are from idiosyncratic DILI patients vs acetaminophen overdose patients.

We thank the reviewer for these comments. We removed the information on idiosyncratic DILI patients from the paper to follow the reviewer's advice and focused our study on patients who overdosed on acetaminophen. Detailed information on patients and healthy subjects has been added in the Materials and Methods section and in Supplementary Table I.

References to the various approved ethics committee studies used in this paper have also been added to this section.

Why was a liver biopsy performed on patients with an acetaminophen overdose? How would it change therapy? How long after the overdose was the biopsy performed?

We thank the reviewer for these comments. We included additional information to clarify the questions. First, the DILI patients had not undergone liver biopsies, but rather, the 13 samples included in the paper were derived from explants of patients undergoing urgent liver transplantation for ALF. This information has been included in the Methods section, under Human samples, and in Supplementary Table I.

Regarding the question related to the timing between overdose and explant, the usual practice for patients with hyperacute liver failure (from paracetamol) is usually within 12 hours. This information has been added to the Method section, under Human samples.

It is stated that "Biological samples from four controls with no liver diseases and tolerant to commonly used drugs attending routine work monitoring were used." In figure 1H there is liver immunohistochemical staining of CNNM4 in 13 acetaminophen overdose patients compared to a healthy group. Presumably the liver sample from the healthy group was obtained by a liver biopsy; I don't know how else material for liver immunohistochemistry could be obtained from the healthy group. A liver biopsy is associated with significant serious risks. It would be very unethical to perform a liver biopsy on a healthy person, and it is hard to believe that it would be part of "routine work monitoring". What type of work? Is there something I am missing?

Regarding the sentence "Biological samples were used from four control subjects who had no liver disease and tolerated the drugs commonly used for routine work monitoring", we added in the revised version of the paper that serum samples were used in this case. Regarding the healthy livers used for the IHC analysis of CNNM4 levels, these were from healthy, non-obese organ transplant donors without liver lesions. This explanation has been added to the Methods section, under Human samples, with the corresponding reference to the ethics committee approval.

The authors seem to equate idiosyncratic DILI with acetaminophen-induced DILI. Acetaminophen DILI is a direct toxin that causes acute toxicity while idiosyncratic DILI is idiosyncratic, delayed in onset, and in most cases mediated by the adaptive immune system, specifically CD8 T cells. The implication is that because CNNM4 decreases acetaminophen DILI in animals it would also work for the treatment of idiosyncratic DILI. It seems very unlikely that

inhibition of CNNM4 would prevent liver injury mediated by CD8 T cells involving perforin and granzymes. I could be wrong, but that hypothesis is not tested in this paper, and it would be difficult to test clinically. Just because serum Mg increases does not mean that CNNM4 is involved; it could just be the result of cell death. Without additional evidence for involvement of CNNM4 in the mechanism of idiosyncratic DILI, I think it is inappropriate to imply that inhibition of CNNM4 would be an effective treatment of this condition.

We thank the reviewer for raising this important issue. We agree with the comment that our results cannot be extrapolated to idiosyncratic DILI. Therefore, we excluded these patients from the paper and included the measurement of magnesium levels from acetaminophen DILI. In this regard, a new panel was added to Figure 1A and the information on these patients was included in the new Supplementary Table I, while the former Supplementary Table I on iDILI patients was removed.

In my opinion, the most important data is the protection by siRNA *Cnnm4*. If I understand correctly, the mice were treated with acetaminophen and 24 hours later they were treated with siRNA, but the of siRNA is not clear. If I understand correctly, the ALT and histology in Figure 5 are 48 hours after the acetaminophen administration and 24 hours after the siRNA administration. I would have liked to see a comparison between the effects of siRNA and glutathione on toxicity since glutathione is the standard therapy for acetaminophen overdoses. I would not expect that glutathione would be very effective 24 hours after the acetaminophen, but I am not sure.

We agree with the reviewer that a comparison with other molecules that previously showed positive results in DILI would be an interesting experiment. Therefore, we treated mice with 360 mg/kg paracetamol and administered N-acetylcysteine (1200 mg/kg) 24 hours later. Animals were sacrificed after 48 hours, and liver damage was assessed by H&E and transaminase levels. We clearly demonstrated that NAC treatment had no beneficial effect on necrotic areas and transaminases in the DILI model. The results were added to the revised version of the paper in Supplementary Figure 5K-L.

The timing in the in vitro experiments is not entirely clear; was the siRNA added at the same time as the acetaminophen? Although I am not an expert in many of the pathways studied in vitro, the data appears to support the basic hypothesis in the case of acetaminophen, and I think that does represent an important contribution.

We thank the reviewer for the comments. In the in vitro experiments, siRNA was added before paracetamol treatment. A more detailed description of the experimental approach has been included in the Supplementary Materials and Methods section. On the advice of other reviewers, we added new data from experiments performed in primary hepatocytes in vitro to understand the importance of CNNM4 as a magnesium effluxer and its effect on the cell death response mediated by APAP overdose, as well as on ER stress and mitochondrial functionality. New results were then included in the following sections:

- (a) Experiments performed with different siRNA sequences (Figure 2B-C)
- (b) Experiments performed with different magnesium concentrations (Figures 2D and 3G and Supplementary Figure 3E)
- (c) Experiments in which primary hepatocytes were transfected with a death mutant of *Cnnm4* T495I for magnesium efflux (Figures 2E, 3E,H, 4C,F and Supplementary Figure 3F-H)

In addition, new experimental approaches to animal models besides NAC treatment were added in the new version of the paper (Supplementary Figure 5K-L):

- (a) Silencing of *Cnnm4* specifically in the liver using a GalNAc formulation in mice treated with APAP for 24 h and sacrificed at 48 h (Figures 5H-M, 6 E-K, and Supplementary Figure 5J)

(b) Silencing of *Cnnm4* specifically in the liver using a GalNAc formulation in mice treated with APAP for 24 h and sacrificed at 36 h (Figure 7D-F and Supplementary Figure 8A-D)

(c) Experiments to substantiate the mechanism underlying the effect shown under *Cnnm4* silencing in AILI animal models (Figures 6A,B,G, Figure 7C, and Supplementary Figures 1A,B, 5B-F, 6B,D, and 7A)

In summary, it seems counter intuitive that a toxin would upregulate a transporter that would increase the toxicity of that toxin, but the authors provide a large amount of data to support that conclusion. I do not believe that the results of studies with acetaminophen can be extrapolated to idiosyncratic DILI. I am very worried about the ethics of the human studies, especially the source of liver samples from normal controls. The native language of the authors is clearly not English, and that makes it more difficult to read; the manuscript requires some editing.

We thank the reviewer for the comments. We have added more detailed information on human studies in the revised version of the paper. In accordance with the reviewer's suggestions, the iDILI results have been deleted. Finally, the paper has been revised by the MDPI English editing service.

tahnk the

Jack Uetrecht

Reviewer #2 (Remarks to the Author):

This study by Gonzales-Recio et al reports that CNNM4 expression is elevated in the liver of acetaminophen-induced liver injury patients and its model mice. CNNM4 is a Mg^{2+} -extruding transporter and its overexpression perturbed mitochondrial Mg^{2+} homeostasis, resulting in augmentation of ROS levels and ER stress. Furthermore, suppression of CNNM4 expression by in vivo delivery of CNNM4 siRNA protected liver from injury by restoring Mg^{2+} homeostasis.

Identification of CNNM4 as a potential therapeutic target for liver injury is intriguing. However, I have several serious concerns regarding the novelty of this study and the experimental validation of the findings.

We thank the reviewer for the suggestions, which improved our knowledge of the role of CNNM4 in APAP DILI. The suggestion regarding *Cnnm4* T495I experiments helped us clarify the importance of Mg^{2+} in liver injury and position CNNM4 as a new therapeutic target. In this new version of the paper: (a) we also included additional information related to Mg^{2+} supplementation; (b) we performed new in vitro experiments with different siRNA sequences and GalNAc molecules to validate the results; (c) we detected enrichment of CNNM4 localization in ER after APAP treatment in animal models, which is also consistent with the information provided by the SubCons webserver, in which a signal peptide responsible for ER localization was identified; (d) high-throughput proteomic analysis revealed, among other things, the importance of *Cnnm4* silencing for the ER stress that occurs during APAP overdose; (e) GalNAc molecules were used to analyze the importance of blocking *Cnnm4* in processes related to mitochondrial dysfunction, ER stress, and liver regeneration in animal models under APAP overdose; (f) a comparative study of the efficacy of N-acetylcysteine, the main therapeutic option in DILI patients, was performed in animal models under APAP treatment compared to *Cnnm4* silencing; and (g) new experimental approaches were used to demonstrate the specificity of IHC staining in the liver and the lack of effect in the intestine.

Finally, in the current paper we removed the information on idiosyncratic DILI patients and added a new cohort of patients who overdosed on paracetamol.

The new experimental data obtained are discussed in the revised version of the paper. The new points uncovered in this study compared to PMID: 33571553 were also highlighted, not only because it is a completely different liver pathology, but also because the underlying mechanism was found to be different from the one previously described. Indeed, the microsomal transfer protein was knocked out to compare the mechanism identified in NAFLD PMID: 33571553 (see results below). No effect was observed in primary hepatocytes under APAP treatment.

Figure. Cell death was evaluated using TUNEL in WT hepatocytes under APAP overdose for 3 h and treated with a siRNA *Cnnm4* and *Mtp*, and compared to a control group (Ctrl). Data are shown as mean \pm SEM. * $P < 0.05$, *** $P < 0.001$ (Student's test)

Major issues:

The same group has reported hepatic CNNM4 overexpression in NASH patients and its model mice (J. Hepatol. 2021, PMID: 33571553). The main conclusion of the NASH paper is very similar to that of this manuscript, but it is not described in this manuscript. The authors should explain the details by citing the previous paper in introduction and discussion, and indicate novel points revealed in this study. In addition, the same paper is actually cited twice as refs. 30 and 37 in different contexts, which should also be corrected appropriately.

We thank the reviewer for the comments. We have added more detailed information on the differences and similarities of process-mediated CNNM4 action in DILI and NAFLD (J. Hepatol. 2021) in the Introduction and Discussion sections of this revised paper.

Although CNNM4 plays an important role in the ER burden in both cases, the mechanism underlying this action is different. While in NAFLD the regulation is mediated by modulation of microsomal transfer protein (MTP) activity, in DILI the observed effects of *Cnnm4* silencing are independent of MTP (see figure above). Proteomics analysis was performed of liver tissue from mice treated with APAP in which *Cnnm4* was silenced after 24 hours of liver injury and sacrificed at 48 hours. The analysis revealed statistical significance in processes related to ER. In addition, cellular localization in the in vivo models under APAP treatment revealed enrichment of CNNM4 in the ER. These and other data cited in the previous commentary have been included in the Results section of paper and added to the Discussion section.

Finally, we corrected the double-cited refs. 30 and 37.

The conclusion heavily depends on RNAi experiments in cultured cells and mice. In general, two or more siRNAs targeting different sequences are needed for validation.

At the suggestion of the reviewer, we have included in the revised version of the paper the following experimental approaches:

(a) Experiments performed with different siRNA sequences (Figure 2B,C)
(b) Experiments performed with different magnesium concentrations (Figures 2D and 3G and Supplementary Figure 3E)
(c) Experiments in which primary hepatocytes were transfected with a death mutant of CNNM4 T495I for magnesium efflux (Figures 2E, 3E,H, 4C,F and Supplementary Figure 3F-H)
In addition, new experimental approaches in animal models besides NAC treatment were added in the new version of the paper (Supplementary Figure 5K-L):

(a) Silencing of *Cnnm4* specifically in the liver using a GalNAc formulation in mice treated with APAP for 24 h and sacrificed at 48 h (Figures 5H-M, 6E-K, and Supplementary Figure 5J)

(b) Silencing of *Cnnm4* specifically in the liver using a GalNAc formulation in mice treated with APAP for 24 h and sacrificed at 36 h (Figure 7D-F and Supplementary Figure 8A-D)

(c) Experiments to substantiate the mechanism underlying the effect shown under *Cnnm4* silencing in ALL animal models (Figures 6A,B,G, and 7C, and Supplementary Figures 1A,B, 5B-F, 6B,D, 7A)

Moreover, definitive conclusions cannot be drawn solely from such type of experiment, which is often associated with non-specific effects. The authors should also take some different approaches (e.g. gene knockout), to firmly establish the importance of CNNM4 in liver injury.

Based on the reviewer's important suggestion, *in vivo* experiments were performed with the GalNAc si*Cnnm4* molecule, kindly provided by Silence Therapeutics, to specifically silence *Cnnm4* in the liver and mainly in hepatocytes. Asialoglycoprotein receptor (ASGPR) PMID: 26000135 is a C-type lectin expressed mainly on the sinusoidal surface of the hepatocyte. Tris-GalNAc (N-acetylgalactosamine) binds to the asialoglycoprotein receptor, which is highly expressed on hepatocytes, resulting in rapid endocytosis. Then GalNAc siRNA conjugates were proposed for delivery to the liver in PMID: 29792572. The experimental design was as described for siRNA *Cnnm4*. Importantly, GalNAc si*Cnnm4* also showed a positive effect on reversing the DILI phenotype in mice treated with APAP overdose, as well as a positive regenerative response to overcome liver injury. These experiments are shown in figures (Figures 5H-M, 6E-K, and 7D-F and Supplementary Figures 5J and 8A-D).

It is difficult to quantitate CNNM4 expression levels by immunohistochemistry. In addition, how did the authors verify the signals? CNNM4 is reportedly localized at the basolateral membrane of the epithelial cells in the intestine (PLOS Genet. 2013, PMID: 24339795), but the staining pattern shown in Supplemental Figure 4B does not seem to coincide with that. Immunofluorescence staining of intestinal sections containing the crypt structure, which are obtained from control and si*Cnnm4*-treated mice (without APAP treatment), will be the best to confirm the authenticity of the signals. Also, the authors should perform immunoblotting experiments for verification and precise quantitation.

The point raised by the reviewer is very important. We have encountered numerous problems in developing a Western blot analysis for the expression of CNNM4 in mouse liver. Indeed, we tested two antibodies: abcam (ab191207) and Mybiosource (MBS9202113). We have added the WB analysis of CNNM4 in Supplementary Figures 1A, 5B,J, and 8B and Figure 6G. For IHC analysis, we performed competitive assays with pure CNNM4 protein to analyze the staining patterns in liver and intestine. These experiments are included in Supplemental Figures 1B and 6D. Detailed information on the technical approach can be found in the Supplementary Materials.

It was previously reported that CNNM4 knockdown results in mitochondrial ROS generation in mammalian cultured cells (Antioxid. Redox Signal. 2020, PMID: 32148064), which seems to be

contradictory to the conclusion of this manuscript. The authors should discuss about the discrepancy.

The reviewer makes a very important point. In the article by Miki et al. (Antioxid. Redox Signal. 2020), the authors show that silencing of *Cnnm4* in the intestine and decreased magnesium levels trigger oxidative stress, ATP depletion, and ROS production in gut epithelial cells. These results suggest that the effect of CNNM4 may vary depending on the metabolic environment in which it is regulated. In this regard, it has been shown that the liver is highly rich in mitochondria compared to other tissues, which gives it a metabolic advantage compared to other cell types. On the other hand, we were able to identify enrichment of CNNM4 in the ER in the liver of mice under APAP treatment (Figure 6G). Blocking the accumulation of CNNM4 in the hepatocytes in the ER under APAP treatment could modulate the interaction of mitochondrial reticulum in these cells and determine their oxidative capacity. A paragraph on this topic has been included in the Discussion section, where we also emphasize the importance of specifically silencing *Cnnm4* in the target tissue to exclude side effects.

Figure 2D, if Mg^{2+} decrease caused by CNNM4 overexpression induces hepatocyte damage, Mg^{2+} supplementation at appropriate levels should restore the damage. Is the addition of 5 mM Mg^{2+} in the medium sufficient? Did the authors check the cytosolic and mitochondrial Mg^{2+} levels? The authors should also test the effect of artificial overexpression of CNNM4 on hepatocyte damage, such as cell death, ROS augmentation, and ER stress. CNNM4 mutants lacking the Mg^{2+} transport activity would be a good negative control.

We thank the reviewer for pointing out these important issues. Regarding the overexpression of *Cnnm4* WT and the *Cnnm4* T495I mutant (pDEST-26 expression plasmid), we have included a detailed characterization of the cell death response, mitochondrial ROS, ER stress, and Mg^{2+} effects in the revised paper (Figures 2E, 3E, H, and 4C,F and Supplementary Figure 3F-H). While overexpression of *Cnnm4* elicits a cell death response and increased ROS and ER stress, the *Cnnm4* T495I mutant did not show the same effect in primary hepatocytes. Mg^{2+} content in mitochondria (Mg-S-TTP-AM) and cytosol (Mg-S-AM) was also measured under *Cnnm4* and *Cnnm4* T495I overexpression. *Cnnm4* overexpression significantly reduced Mg^{2+} content in mitochondria, whereas *Cnnm4* T495I expression completely abolished Mg^{2+} efflux in hepatocytes. Importantly, treatment with 5, 10, 20, and 40 mM Mg^{2+} failed to counteract the damage caused by APAP (Supplementary Figure 3E) or *Cnnm4* overexpression, as measured by TUNEL and annexin V apoptosis and necrosis assays (Figure 2D). Mg^{2+} content was also measured in the presence of 5 and 20 mM of this cation under APAP treatment (Figure 3G). Mg^{2+} content in mitochondria was not restored under these conditions (Figure 3G), likely due to the upregulation of CNNM4 induced by the hepatotoxic agent.

In conclusion, these data suggest that the effect observed after silencing of *Cnnm4* in primary hepatocytes under APAP treatment is mainly mediated by its activity as an Mg^{2+} effluxer. Moreover, Mg^{2+} supplementation does not counteract the reduction of this cation observed under APAP treatment or overexpression of *Cnnm4*. The significance of these results has been included in the Discussion section.

Figure 3A, Mg^{2+} levels are strongly decreased in the mitochondria. This is very curious because CNNM4 proteins are supposedly localized at the plasma membrane. Is there any mechanism to maintain cytosolic Mg^{2+} levels at the expense of mitochondrial Mg^{2+} ? Alternatively, CNNM4 proteins may be partly localized at the mitochondrial membrane and regulate Mg^{2+} homeostasis in the mitochondria, as illustrated in Figure 3. The authors should address this intriguing problem by experimentation.

This is a very interesting point raised by the reviewer. By subcellular fractionation, we found a statistical enrichment of CNNM4 in the ER of livers of animals under APAP overdose, also consistent with the information provided by the SubCons webserver, in which a signal peptide

responsible for ER localization was identified. The mitochondrial fraction did not exhibit CNNM4, whereas the amount of this protein in the membrane remained constant (Figure 6G). MAMs are interaction zones between mitochondria and ER, allowing rapid exchange of molecules, including Ca^{2+} and possibly Mg^{2+} , to maintain cellular function and health. Therefore, the presence of CNNM4 in ER under APAP toxicity likely disrupts Mg^{2+} homeostasis between the two organelles and triggers a misfolding protein response and mitochondrial ROS under these conditions. These data have been added to the Discussion section.

Minor issues:

Figure 4A, the data are not clear with large variations and seem to be heavily modified by image-processing software.

Following the suggestion of the reviewer, we have modified Figure 4A; in this revised version it is Figure 4B.

Figure 4C, what is the meaning of S6 phosphorylation?

We removed S6 phosphorylation in the revised version of the paper.

Figure 4D, some explanation is needed for Ca^{2+} oscillation by thapsigargin treatment, by citing appropriate papers.

Thapsigargin is a simple tool to deplete stores by inhibiting the sarco-endoplasmic reticulum Ca^{2+} -ATPase (SERCA) pump (Thastrup O, Dawson AP, Scharff O, Foder B, Cullen PJ, Drøbak BK, Bjerrum PJ, Christensen SB, Hanley MR. Thapsigargin, a novel molecular probe for studying intracellular calcium release and storage. *Agents Actions*. 1989 Apr;27(1-2):17-23. doi: 10.1007/BF02222186. PMID: 2787587). Here, Ca^{2+} oscillation induced by thapsigargin occurs because during the first 200 s of Ca^{2+} kinetics determination, measurement is performed in Ca^{2+} -free solution and reflects only Ca^{2+} release from the stores. Adding Ca^{2+} extracellularly allows readmission of external Ca^{2+} through store-operated Ca^{2+} influx. Similar results can be found in Lin, YP., Bakowski, D., Mirams, G.R. et al. Selective recruitment of different Ca^{2+} -dependent transcription factors by STIM1-Orai1 channel clusters. *Nat Commun* 10, 2516 (2019).

Supplemental Figure 1C, lines connecting each data point should be removed.

We removed the lines in Supplemental Figure 1C; now it is Supplementary Figure 2C.

Supplemental Figure 4B, CNNM4 expression levels are greatly reduced by APAP treatment, but why? Also, the non-effectivity of siCnnm4 may be due to this decrease.

In Supplementary Figure 4B, now Supplementary Figure 6A, CNNM4 levels were measured in the intestine. The expression of CNNM4 was drastically reduced by APAP treatment in both control and *Cnnm4* siRNA experimental groups without significant differences. Also, knockdown of *Cnnm4* in healthy mice did not change the expression in the intestine (Supplementary Figure 6B). These results exclude non-hepatic sites of action for *Cnnm4* silencing.

Supplemental Figure 4C, is there any statistical difference between “APAP + siCtrl” and “APAP + siCnnm4”?

In the revised version of the paper, this is now Supplementary Figure 6C. We reanalyzed the statistical differences between APAP + siCtrl and APAP + si*Cnnm4*m and the *P*-value is 0.06. Scale bars in microscopic images are too small in many figures.

We modified the scale bars in the microscopic images.

Reviewer #3 (Remarks to the Author):

Authors investigated the role of transporters family Cyclin M4, also known as CNNM4 that is a key role in the Mg²⁺ transport across the cell membranes, in acetaminophen (APAP)-induced liver injury. They showed increased expression of CNNM4 in APAP-treated cultured hepatocytes, mouse livers and liver samples of patients with APAP-induced liver injury (ALI), and these patients also had increased serum Mg²⁺ levels. More importantly, silencing CNNM4 in the liver with nanoparticle- and GalNAc-formulated siRNA protected against liver from ALI. While to study Mg²⁺ in APAP is novel and the study has clinical and translation value, there were major concerns over the lack of rigor and robustness of this study. The mechanistic parts were also weak and not sure whether ER stress would be the primary mechanism to explain the protective effects of CNNM4 knockdown.

We thank the reviewer for the interesting questions and suggestions, which have undoubtedly improved our paper. The issue of treatment of DILI disease is exactly what we would like to highlight in this paper in the context of CNNM4 regulation. Possible intervention in DILI by modulating ER stress has already been reported in several publications (PMID: 32790969, PMID: 23665281, PMID: 27470132). In our case, we added new experiments to this version of the paper and developed in vivo experiments with mice treated with GalNAc CNNM4 formulation and APAP, which brings this treatment closer to the clinical picture. On the other hand, we designed experiments with tunicamycin, an ER stress trigger, to see the direct effect of CNNM4, and we verified the efficacy of its removal. Finally, to learn more about the mechanism underlying CNNM4 silencing, we performed experiments with the CNNM4 T495I mutant, which loses the ability to efflux magnesium, to determine whether it is this activity that regulates ER stress and mitochondrial dysfunction. Importantly, we found that there was an accumulation of CNNM4 in the ER in livers of mice under APAP treatment. Finally, we performed high-throughput proteomics on the livers of CNNM4-silenced mice treated and not treated with APAP to better understand the pathophysiological mechanism leading to CNNM4 modulation, confirming pathways related to ER activity.

The in vitro studies were performed after 1-6 hrs treatment of APAP (which would be considered as early event), and in vivo APAP model was 48hrs post treatment (considered as late event). It is unclear how the mechanisms studied in vitro relevant to the in vivo mouse studies? If ER stress is key mechanism downstream of Cnnm4 knockdown, then targeting ER stress per se should also have beneficial effects for this APAP injury mouse model?

The reviewer makes an interesting point regarding in vitro and in vivo studies. Lengthy in vitro studies with primary hepatocytes are difficult to perform because of the differentiation of hepatocytes in culture. On the advice of other reviewers, we added new experimental approaches performed on primary hepatocytes in vitro to understand the importance of CNNM4 as a magnesium effluxer and its effect on the cell death response mediated by APAP overdose, as well as on ER stress and mitochondrial functionality. New data were then included in the following sections:

- (a) Experiments performed with different siRNA sequences (Figure 2B,C)
 - (b) Experiments performed with different magnesium concentrations (Figures 2D and 3G and Supplementary Figure 3E)
 - (c) Experiments in which primary hepatocytes were transfected with a death mutant of CNNM4 T495I for magnesium efflux (Figures 2E, 3E,H, and 4C,F and Supplementary Figure 3F-H)
- In addition, new experimental approaches in animal models besides NAC treatment were added in the new version of the paper (Supplementary Figure 5K-L):

(a) Silencing of *Cnnm4* specifically in the liver using a GalNAc formulation in mice treated with APAP for 24 h and sacrificed at 48 h (Figures 5H-M and 6E-K and Supplementary Figure 5J)

(b) Silencing of *Cnnm4* specifically in the liver using a GalNAc formulation in mice treated with APAP for 24 h and sacrificed at 36 h (Figure 7D-F and Supplementary Figure 8A-D)

(c) Experiments to substantiate the mechanism underlying the effect shown under CNNM4 silencing in ALLI animal models in vivo (Figures 6A,B,G and 7C-and Supplementary Figures 1A,B, 5B-F, 6B,D, and 7A)

The results of experimental approaches performed in vitro, including the use of CNNM4 T495I and the addition of Mg^{2+} , confirmed that the effects of CNNM4 are Mg^{2+} -dependent and mediated by ER and mitochondrial activity. These results were validated in vivo, showing an enrichment of CNNM4 in ER after APAP treatment. Furthermore, silencing of *Cnnm4* in vivo reversed APAP-induced damage and enhanced liver regenerative capacity, likely due in part to high ATP production.

Silencing of *Cnnm4* opens a therapeutic window for DILI that remains undiscovered in clinical practice.

Major concerns:

1. There were concerns over the data rigor and robustness. There was lacking of experimental details in the figure legend, and many animal experimental data were only from 3 mice. For instance, for many experiments, the dose of APAP and whether the mice were fasted or not were not described.

We thank the reviewer for the comments. We added details in the figure legends and included two new in vivo experiments. First, animals were treated with *Cnnm4* GalNAc siRNA molecules and APAP to bring further approaches to silencing *Cnnm4* closer to clinical practice. The results of silencing *Cnnm4* specifically in the liver using a GalNAc formulation in mice treated with APAP for 24 hours and sacrificed at 36 or 48 hours were included (Figures 5H-M, 6E-K, and 7D-F and Supplementary Figures 5J and 8A-D).

A second group of mice was treated with NAC for 24 hours to investigate its efficacy in APAP overdose, and it was found that NAC was not effective in this time window (Supplementary Figure 5K-L). We also included more animals in the in vivo experiments and increased the number of in vitro experiments shown in the paper to add robustness to the data.

Regarding the reviewer's question about whether the animals were fasted or starved, we have included this information in the Supplementary Material and Methods section, which describes that the animals were starved for 12 hours.

2. Figure 1, one the most key piece of data presented in Figure 1A were from 10 mixed DILI patients. While this may tend to suggest that increased serum Mg^{2+} may be a general event, it will be nice to show how specific this to APAP-induced liver injury. Authors may consider to obtaining more samples just from APAP liver injury patients to specifically address its correlation with APAP.

We thank the reviewer for raising this important question. We agree with the comment that our findings could not be extrapolated to idiosyncratic DILI, therefore we have eliminated these patients from the paper and incorporated the measurement of magnesium levels from acetaminophen DILI. A new panel in Figure 1A has been added in this respect and the information on these patients has been added to the new Supplementary Table 1, and the previous Table 1 related to iDILI patients has been removed.

3. Figure 1, in addition to mRNA and IHC staining, the increased CNNM4 should be confirmed by western blot analysis (which will be more quantitative at the protein levels). In addition, it will be more insightful if they authors can show where are the cellular locations of CNNM4 after APAP treatment.

The point raised by the reviewer is very important. We encountered numerous problems in developing a Western blot analysis for the expression of CNNM4 in mouse liver. Indeed, we tested two antibodies: abcam (ab191207) and Mybiosource (MBS9202113). We added the WB analysis of CNNM4 in Supplementary Figure 1A and Figures 5B,J, 6G, and 8B. For IHC analysis, we performed competitive assays with pure CNNM4 protein to analyze the staining pattern in liver and intestine. These experiments are included in Supplementary Figures 1B and 6D. Detailed information on the technical procedure can be found in the Supplementary Materials section.

Regarding the cellular localization of CNNM4, we found by subcellular fractionation statistical enrichment of CNNM4 in the ER of liver of animals under APAP overdose, which is also in agreement with the information provided by the SubCons webserver, in which a signal peptide responsible for the ER localization was identified. The mitochondrial fraction did not exhibit CNNM4, while the amount of this protein in the membrane remained constant (Figure 6G). These experiments have been added to the Discussion section.

4. Figure 2, what were the concentrations of APAP used in these cell cultures studies? How the TUNEL positive cells were quantified? More robust and objective assays such as LDH leakage should be considered. More importantly, the knockdown and overexpression efficiency of Cnnm4 should be validated. For all in vitro studies, the APAP metabolism and bioactivation after Cnnm4 knockdown should be determined as this is critical to rule out its effects on APAP metabolism.

We thank the reviewer for these comments. The concentrations of APAP used in cell cultures studies were included in the Supplemental Materials, along with the TUNEL analysis used for positive cell quantification. Additionally, we have incorporated a detailed characterization of the apoptotic response in the revised paper; apoptosis and necrosis were also analyzed by RealTime Glo annexin V assay (Figure 2A-E and Supplementary Figure 3A-G).

Finally, we have added CYP2E1 activity in the in vitro experiments under *Cnnm4* silencing and CNNM4 and CNNM4 T495I overexpression (Supplementary Figure 3H). Our results confirm the data previously reported by others showing that CYP2E1 activity was reduced in cells treated with APAP. Importantly, *Cnnm4* silencing induces this activity in primary hepatocytes, while T495I mutant did not exert any effect.

5. Figure 4, in addition to the mRNA changes of the UPR, some of them should be confirmed at the protein levels.

Following the suggestions of the reviewer, we have added results related to ER stress targets by Western blot analysis in Figure 6F.

6. Figure 5, the si Cnnm4 alone group was missing. More importantly, the knockdown efficiency of Cnnm4 in the mouse liver should be validated by western blot analysis.

The point raised by the reviewer is very important. We have encountered numerous problems in developing a Western blot analysis for the expression of CNNM4 in mouse liver. Indeed, we tested two antibodies: abcam (ab191207) and Mybiosource (MBS9202113). We have added the WB analysis of CNNM4 in Supplementary Figure 1A and Figures 5B, J, 6G, and 8B. For IHC analysis, we performed competitive assays with pure CNNM4 protein to analyze the staining

patterns in liver and intestine. These experiments are included in Supplementary Figures 1B and 6D. Detailed information about the technical approach can be found in the Supplementary Materials section.

In addition, we also added H&E images, transaminases levels, F4/80, and magnesium levels in control mice with *Cnnm4* silenced (Supplementary Figure 5C-F). Inhibition of CNNM4 in the basal group had no effect, except on serum magnesium level, which was significantly reduced.

More detail mechanisms were missing in this important part of in vivo work on how knockdown of *Cnnm4* protected against APAP liver injury. Previously showed mitochondrial damage should be examined in the in vivo study to highlight the relevance of their in vitro findings. As this model is post APAP 48 hrs, it is known at this time point the mouse has passed the injury phase and most likely enter the recovery phase.

Following the reviewer's suggestion, we performed additional in vivo experiments with GalNAC CNNM4 to find a new approach for silencing CNNM4 closer to clinical practice (Figure 6H-K). We examined mitochondrial respiration by seahorse analysis in purified liver mitochondria, mitochondrial ROS, and ATP in hepatocytes from mice treated with APAP and APAP GalNAC CNNM4. We also analyzed ER staining under these experimental conditions. CNNM4 accumulated in the ER under APAP treatment (Figure 6G), and proteomics analysis of APAP and/or silenced CNNM4 livers (Figure 6A-B), where GO overrepresentation was linked to ER activity, confirmed the in vivo results found in vitro.

The impact of *cnnm4* knockdown on increased liver regeneration (PCNA staining) is interesting but how knockdown of *cnnm4* increase cell proliferation was not determined in detail. 7. Figure 6, the caspase 3 activity and apoptosis data interpretation was questionable as it is well known APAP induced necrosis and caspase activation is not relevant in this model. 8. It was unclear why the manganese superoxide dismutase activities after manipulating *cnnm4* was not determined? Which may be an important protective mechanism against APAP liver injury of *cnnm4* knockdown. It was also unclear how these inflammatory gene expression changes would be mechanistically linked to hepatocytes proliferation and regeneration.

Following the reviewer's suggestion, we performed the TUNEL assay on liver parenchyma sections from animals treated with APAP or animals with silenced *Cnnm4* with si*Cnnm4* (Figure 5C) or GalNAc siRNA molecule (Figure 5J). The analysis showed a significant reduction in cell death under *Cnnm4* silencing.

To understand how silencing *Cnnm4* could trigger liver regeneration, the upstream factors involved in these processes are examined in detail in Figure 7C,D. HGF was significantly upregulated when CNNM4 was absent. In correlation, phosphorylation of c-Met appeared to be activated under CNNM4 silencing (Figure 7F) and correlated with downregulation of p21, a cell cycle checkpoint (Figure 7E). p21 was directly associated with impaired liver regeneration observed after severe APAP-induced liver injury in patients (PMID: 29540258). This reduction in p21 along with an increase in cyclin D1 and PCNA levels as well as p-c-Met suggests that the absence of CNNM4 enhances the regenerative response in ALF. These data and the link to cellular ATP levels essential for liver regeneration were added to the Discussion section.

Reviewer #4 (Remarks to the Author):

My expertise is limited to the siRNA therapeutics and not liver injury. I wish to comment solely on the design of the RNA molecule. It is not immediately obvious if sequence design and stability

was considered. These are likely downstream considerations of the present study, but are certainly worth a short discussion.

siRNAs were designed in silico using proprietary algorithm. A limited number of molecules were synthesized and tested experimentally for activity, specificity and safety. On target activity and potential off target gene regulation of lead candidates were assessed by transcriptome analyses (RNAseq) of primary hepatocytes exposed to GalNAc conjugates prior to *in vivo* testing.

We added more details in the Methods section to the chemistry of siRNA conjugates which were designed for enhanced stability which is essential for in vivo application.

REVIEWER COMMENTS

Reviewer #1 (Remarks to the Author):

The manuscript is greatly improved relative to the original submission. Some details are much clearer, although there are still phrases such as “treated with APAP 360 mg/kg for 48 hours”. I assume that this means that the APAP was administered in a single dose, and then samples were taken 48 hours later rather than APAP was administered slowly over a period of 48 hours as the sentence implies. Sentences such as “The death mutant Cnm4-T4951 showed no effect on cell death.” is confusing. Does that mean that APAP did not cause cell death in Cnm4-T4951 cells?

I still have one major problem with the work. The potential importance of the work is two-fold: it could provide additional information about the mechanism of APAP liver injury, and it could provide a potential therapy 48 hours after the overdose. However, there is a problem with the timing. In the few studies that we have done with APAP, the ALT was markedly elevated at 24 hours, but almost back to normal by 48 hours. In this manuscript, there is no measurement at 24 hours, but the ALT was markedly elevated at 48 hours. In fact, the ALT was higher than we found at 24 hours. There are differences in the methods of our studies, e.g., we used a dose of 300 mg/kg rather than 360 mg, but I don't think that explains the timing. Although it is possible that there are other mechanisms at play, ALT is usually considered evidence of cell necrosis, and I think it is clear that the injury caused by APAP is quite acute. It is likely that most of the cell death occurs well before 48 hours; therefore, how could siCnm4 have an effect at 48 hours? In the Methods section, in the study of APAP overdose patients, the timing between the APAP overdose and the explant was “usually within 12 hours”. If the liver is so damaged at 12 hours that a transplant is required, how could treatment at 24 hours be of benefit. It is conceivable that the treatment helps liver regeneration rather than preventing liver damage, but there is no evidence to support that hypothesis.

Reviewer #2 (Remarks to the Author):

The authors have adequately addressed most of the comments on the previous version and revised the manuscript. However, there still remains one crucially important issue that has left unaddressed in this revision.

CNNM4 upregulation by APAP overdose is a key finding of this study, and clear demonstration of increase in CNNM4 protein levels is a crucial point for evaluating this manuscript. The authors added results of CNNM4-immunoblotting analyses, but there are several concerns about the data. For quantitative immunoblotting analyses, it is necessary to compare signal intensities on the same membrane, but the data presented in the manuscript are all pieced together. Appropriate comparison cannot be done based on these fragmentary results. Also, how did the authors verify the authenticity of the signal? In the reply to another reviewer (Reviewer #3), the authors describe the difficulty in

detecting CNNM4 by immunoblotting analyses using two commercially available antibodies. Was the band specific to the part of the target protein size, yielding virtually only single band on the full blot? If there were non-specific bands around it, it should be shown that only the target band signal is increasing with APAP treatment or decreasing with CNNM4-KD. Full blot images would be desirable.

Regarding the immunostaining of the intestine, I requested to perform staining of tissue sections containing the crypt structure in the previous comment. However, no correction has been made on this point. In the intestine, CNNM4 has been shown to localize to the basolateral membrane of epithelial cells in the mucosal layer, which is reliable because the signal disappears in CNNM4-KO mice, making it suitable for evaluating the authors' immunostaining experiments based on the pattern of signals. The authors performed competition experiments with pure CNNM4 proteins instead, but I could not find any data or method description of how the pure CNNM4 protein was prepared or how the quality of the protein was verified. Competition experiments with some equivalent proteins lacking the antigenic moiety, which were prepared by the same procedure, should be performed as a control experiment, but this has not been done either. Experiments lacking these basic information and adequate controls are difficult to evaluate.

Reviewer #3 (Remarks to the Author):

Authors have performed additional experiments and most of my concerns have been addressed. The quality of this manuscript has been improved.

However some minor issues remained.

Figure 6G for the membrane, mitochondria and ER fraction blots on the same blot? authors should provide multiple mouse fractionation samples and run on the same blot to make the data more robust and rigor.

RESPONSE: We thank the Reviewers for the positive view of our manuscript and the excellent comments and suggestions. A detailed point-by-point response to the comments is included below.

Reviewer #1 (Remarks to the Author):

The manuscript is greatly improved relative to the original submission. Some details are much clearer, although there are still phrases such as “treated with APAP 360 mg/kg for 48 hours”. I assume that this means that the APAP was administered in a single dose, and then samples were taken 48 hours later rather than APAP was administered slowly over a period of 48 hours as the sentence implies. Sentences such as “The death mutant Cnm4-T4951 showed no effect on cell death.” is confusing. Does that mean that APAP did not cause cell death in Cnm4-T4951 cells?

RESPONSE: We totally agree with the Reviewer and accordingly we have modified the sentences and replaced them. First, “treated with APAP 360 mg/kg for 48 hours” was replaced by “APAP was administered in a single dose of 360 mg/kg, and 48 hours later, mice were sacrificed” in the MS. Regarding “The death mutant Cnm4-T4951 showed no effect on cell death.” We have modified this sentence to “Cnm4-T4951 showed a different effect of Cnm4 wild type overexpression”.

I still have one major problem with the work. The potential importance of the work is two-fold: it could provide additional information about the mechanism of APAP liver injury, and it could provide a potential therapy 48 hours after the overdose. However, there is a problem with the timing. In the few studies that we have done with APAP, the ALT was markedly elevated at 24 hours, but almost back to normal by 48 hours. In this manuscript, there is no measurement at 24 hours, but the ALT was markedly elevated at 48 hours. In fact, the ALT was higher than we found at 24 hours. There are differences in the methods of our studies, e.g., we used a dose of 300 mg/kg rather than 360 mg, but I don't think that explains the timing. Although it is possible that there are other mechanisms at play, ALT is usually considered evidence of cell necrosis, and I think it is clear that the injury caused by APAP is quite acute. It is likely that most of the cell death occurs well before 48 hours; therefore, how could siCnm4 have an effect at 48 hours? In the Methods section, in the study of APAP overdose patients, the timing between the APAP overdose and the explant was “usually within 12 hours”. If the liver is so damaged at 12 hours that a transplant is required, how could treatment at 24 hours be of benefit. It is conceivable that the treatment helps liver regeneration rather than preventing liver damage, but there is no evidence to support that hypothesis.

RESPONSE: The reviewer raises another very important question.

First, we made an error regarding the timing between APAP overdose and explant in patients, and we thank the reviewer for this important comment. We have included the specific time for the explants of these patients, which was approximately 3 days. These data are added in Supplemental Table II. Therefore, the therapeutic window for GalNAc Cnm4 will be consistent with clinical practice.

Regarding the timing, as the reviewer notes, we have performed our experiments as described in the Materials and Methods section by silencing CNNM4 after 24 hours of APAP injection (360 mg/Kg) and sacrificed the animals 24 hours later (48 hours after APAP injection). To determine the efficacy of CNNM4 silencing during that 24 hours, we have analyzed the hepatic mRNA levels of *Cnnm4* at different time points in that interval of time. The results are shown in the figure below. We found that *Cnnm4* levels are reduced 6 hours after treatment with GalNAc (30h after APAP overdose). *Cnnm4* is also effectively silenced within 12 hours and 24 hours, corresponding to 36 and 48 h of APAP treatment respectively, which could exert its protective effect.

Figure. mRNA expression levels of *Cnnm4* in WT mice treated with APAP 360 mg/kg. 24 hours after APAP overdose, mice were silenced with GalNAc si*Cnnm4* or siCtrl for 6 hours, 12 hours and 24 hours. Mice were treated with APAP overdose for a total of 30 hours, 36 hours and 48 hours, respectively. Values are represented as mean \pm SEM. ** P < 0.01, ***P < 0.001 (Student's test)

Finally, we fully agree with the reviewer that treatment with GalNAc *Cnnm4* could also support liver regeneration, as shown in Figure 7 and Supplementary Figure 8. Mitochondrial and ER activity and ATP production could also contribute to liver regeneration, as described by our group Goikoetxea-Usandizaga N Hepatology. 2022 Mar;75(3):550-566 and Barbier-Torres L Nat Commun. 2017 Dec 12;8(1):2068.

As a summary, the new results included in the following sections are:

- (a) Western blot of CNNM4 of preclinical animal models treated with a single dose of APAP 360mg/kg and 12h and 24h thereafter mice were silenced with si*Cnnm4*, GalNAc si*Cnnm4* or an unrelated control. Mice were sacrificed 36h and 48 after APAP treatment, respectively (Figures 5A, 5H, Supplementary Figure 6B, Supplementary Figure 7A and Supplementary Figure 10B)
- (b) IHC analysis in liver tissue of preclinical animal models treated with a single dose of APAP 360mg/kg and 12h and 24h thereafter mice were silenced with si*Cnnm4*, GalNAc si*Cnnm4* or an unrelated control. Mice were sacrificed 36h and 48h after APAP treatment, respectively (Supplementary Figure 6C, Supplementary Figure 7B and Supplementary Figure 10C)
- (c) Western blot of CNNM4 location in membrane, mitochondria and endoplasmic reticulum in preclinical animal models treated with a single dose of APAP 360mg/kg for 48 hours and compared with a control group (Figure 6G)

- (d) Competitive assays with pure CNNM4 protein (CNNM4₃₅₆₋₇₇₅) and CNNM4 protein without the antigenic moiety to analyze the staining pattern in crypt sections of intestine. (Supplementary Figure 8D). Detailed information of the protein purification procedure can be found in the Supplementary Materials section and a blot with quality of the protein in the Supplementary Figure 1B.
- (e) Experiments of IHC staining of CNNM4 in the crypt structure of the intestine in preclinical animal models treated with a single dose of APAP 360mg/kg and 24 hours later mice were silenced with *siCnnm4*. Mice were sacrificed 48h after APAP treatment (Supplementary Figure 8A)
- (f) Uncropped blots were added in the Supplementary Figure 11 and 12.

Reviewer #2 (Remarks to the Author):

The authors have adequately addressed most of the comments on the previous version and revised the manuscript. However, there still remains one crucially important issue that has left unaddressed in this revision.

CNNM4 upregulation by APAP overdose is a key finding of this study, and clear demonstration of increase in CNNM4 protein levels is a crucial point for evaluating this manuscript. The authors added results of CNNM4-immunoblotting analyses, but there are several concerns about the data. For quantitative immunoblotting analyses, it is necessary to compare signal intensities on the same membrane, but the data presented in the manuscript are all pieced together. Appropriate comparison cannot be done based on these fragmentary results.

RESPONSE: We thank the reviewer for these comments. We repeated the CNNM4-immunoblotting analyses in the same membrane to compare the signal intensities. We added the corresponding Western blot in the Figure 5A, Figure 5H, Figure 6G, Supplementary Figure 2A, Supplementary Figure 6B, Supplementary Figure 7A and Supplementary Figure 10B. Moreover, we performed IHC analysis in liver tissue of preclinical animal models treated with a single dose of APAP 360mg/kg for and 12h and 24h later mice were silenced with *siCnnm4*, GalNAc *siCnnm4* or an unrelated control. Mice were sacrificed 36h and 48h, respectively. We added the corresponding immunostaining analysis in the Supplementary Figure 6C, Supplementary Figure 7B and Supplementary Figure 10C.

Also, how did the authors verify the authenticity of the signal? In the reply to another reviewer (Reviewer #3), the authors describe the difficulty in detecting CNNM4 by immunoblotting analyses using two commercially available antibodies. Was the band specific to the part of the target protein size, yielding virtually only single band on the full blot? If there were non-specific bands around it, it should be shown that only the target band signal is increasing with APAP treatment or decreasing with CNNM4-KD. Full blot images would be desirable.

RESPONSE:

We thank the reviewer for these comments. Uncropped blots were added in the Supplementary Figure 11 and 12. To verify the authenticity of the signal, we performed a competitive analysis with the pure protein in the Western blot membrane. We found that the signal disappeared in this competitive assay, confirming the specificity of the CNNM4 band. The result of this competitive analysis is shown in the following figure:

Figure. Competitive assay of Cnnm4 signal with a pure protein CNNM4^{BATEMAN-cNMP-Ctail.(356-775)} and antiCNNM4 antibody (ab191207) in the Western Blot membrane.

Regarding the immunostaining of the intestine, I requested to perform staining of tissue sections containing the crypt structure in the previous comment. However, no correction has been made on this point. In the intestine, CNNM4 has been shown to localize to the basolateral membrane of epithelial cells in the mucosal layer, which is reliable because the signal disappears in CNNM4-KO mice, making it suitable for evaluating the authors' immunostaining experiments based on the pattern of signals.

RESPONSE:

We fully agree with the reviewer and accordingly modified the immunostaining of CNNM4 in the intestine and added a new analysis with tissue sections containing the crypt structure. This new result is shown in supplemental Figure 8A.

We can observe that the expression of CNNM4 increased in the crypt structure by APAP treatment in both control and Cnnm4 siRNA experimental groups, with no significant differences.

The authors performed competition experiments with pure CNNM4 proteins instead, but I could not find any data or method description of how the pure CNNM4 protein was prepared or how the quality of the protein was verified. Competition experiments with some equivalent proteins lacking the antigenic moiety, which were prepared by the same procedure, should be performed as a control experiment, but this has not been done either. Experiments lacking these basic information and adequate controls are difficult to evaluate

RESPONSE: We thank the reviewer for these comments. We have included the method description of the protein purification in the Supplemental material in the section "Expression and purification of recombinant CNNM4^{BATEMAN-cNMP-Ctail.(356-775)}" and a blot with quality of the protein in the Supplementary Figure 1A. Moreover, we have performed a competition experiment with the same protein lacking the antigen moiety CNNM4^{cNMP (545-730)} included in Supplementary Figure 8D. We added the method description of the protein purification in the Supplementary material in the section "Expression and purification of recombinant CNNM4^{cNMP (545-730)}" and a blot with quality of the protein in the Supplementary Figure 1B.

As a summary, the new results included in the following sections are:

- (a) Western blot of CNNM4 of preclinical animal models treated with a single dose of APAP 360mg/kg and 12h and 24h thereafter mice were silenced with siCnnm4, GalNAc siCnnm4 or an unrelated control. Mice were sacrificed 36h and 48 after APAP treatment, respectively (Figures 5A, 5H, Supplementary Figure 6B, Supplementary Figure 7A and Supplementary Figure 10B)

- (b) IHC analysis in liver tissue of preclinical animal models treated with a single dose of APAP 360mg/kg and 12h and 24h thereafter mice were silenced with *siCnnm4*, GalNAc *siCnnm4* or an unrelated control. Mice were sacrificed 36h and 48h after APAP treatment, respectively (Supplementary Figure 6C, Supplementary Figure 7B and Supplementary Figure 10C)
- (c) Western blot of CNNM4 location in membrane, mitochondria and endoplasmic reticulum in preclinical animal models treated with a single dose of APAP 360mg/kg for 48 hours and compared with a control group (Figure 6G)
- (d) Competitive assays with pure CNNM4 protein (CNNM4₃₅₆₋₇₇₅) and CNNM4 protein without the antigenic moiety to analyze the staining pattern in crypt sections of intestine. (Supplementary Figure 8D). Detailed information of the protein purification procedure can be found in the Supplementary Materials section and a blot with quality of the protein in the Supplementary Figure 1B.
- (e) Experiments of IHC staining of CNNM4 in the crypt structure of the intestine in preclinical animal models treated with a single dose of APAP 360mg/kg and 24 hours later mice were silenced with *siCnnm4*. Mice were sacrificed 48h after APAP treatment (Supplementary Figure 8A)
- (f) Uncropped blots were added in the Supplementary Figure 11 and 12.

Reviewer #3 (Remarks to the Author):

Authors have performed additional experiments and most of my concerns have been addressed. The quality of this manuscript has been improved.

However some minor issues remained.

Figure 6G for the membrane, mitochondria and ER fraction blots on the same blot? authors should provide multiple mouse fractionation samples and run on the same blot to make the data more robust and rigor.

RESPONSE: We thank the reviewer for these comments. We repeated the CNNM4-immunoblotting analyses in the same membrane to compare the signal intensities. We added the corresponding Western blot in the Figure 6G.

As a summary, the new results included in the following sections are:

- (a) Western blot of CNNM4 of preclinical animal models treated with a single dose of APAP 360mg/kg and 12h and 24h thereafter mice were silenced with *siCnnm4*, GalNAc *siCnnm4* or an unrelated control. Mice were sacrificed 36h and 48 after APAP treatment, respectively (Figures 5A, 5H, Supplementary Figure 6B, Supplementary Figure 7A and Supplementary Figure 10B)
- (b) IHC analysis in liver tissue of preclinical animal models treated with a single dose of APAP 360mg/kg and 12h and 24h thereafter mice were silenced with *siCnnm4*, GalNAc *siCnnm4* or an unrelated control. Mice were sacrificed 36h and 48h after APAP treatment, respectively (Supplementary Figure 6C, Supplementary Figure 7B and Supplementary Figure 10C)
- (c) Western blot of CNNM4 location in membrane, mitochondria and endoplasmic reticulum in preclinical animal models treated with a single dose of APAP 360mg/kg for 48 hours and compared with a control group (Figure 6G)
- (d) Competitive assays with pure CNNM4 protein (CNNM4₃₅₆₋₇₇₅) and CNNM4 protein without the antigenic moiety to analyze the staining pattern in crypt sections of intestine. (Supplementary Figure 8D). Detailed information of the

protein purification procedure can be found in the Supplementary Materials section and a blot with quality of the protein in the Supplementary Figure 1B.

- (e) Experiments of IHC staining of CNNM4 in the crypt structure of the intestine in preclinical animal models treated with a single dose of APAP 360mg/kg and 24 hours later mice were silenced with *siCnnm4*. Mice were sacrificed 48h after APAP treatment (Supplementary Figure 8A)
- (f) Uncropped blots were added in the Supplementary Figure 11 and 12.

REVIEWERS' COMMENTS

Reviewer #1 (Remarks to the Author):

This manuscript has come a long way since the first version. I am still skeptical that the proposed mechanism will allow for late treatment of acetaminophen overdoses, but the data are interesting, and they should provide a stimulus for further studies. In short, I recommend publication.

Reviewer #2 (Remarks to the Author):

The authors have satisfactorily addressed the problems, and the revised manuscript has been greatly improved. I have no further comments.

Reviewer #3 (Remarks to the Author):

I am satisfied with the revised version.

REVIEWERS' COMMENTS

Reviewer #1 (Remarks to the Author):

This manuscript has come a long way since the first version. I am still skeptical that the proposed mechanism will allow for late treatment of acetaminophen overdoses, but the data are interesting, and they should provide a stimulus for further studies. In short, I recommend publication.

We thank the reviewer for this question. We have included in the discussion the timing of the therapy. "These results pave the way for CNNM4 to be considered as a potential therapeutic target for APAP overdose conditions lasting 6 to 24 hours. Our findings suggested a 24-hour timing therapy."

Reviewer #2 (Remarks to the Author):

The authors have satisfactorily addressed the problems, and the revised manuscript has been greatly improved. I have no further comments.

Reviewer #3 (Remarks to the Author):

I am satisfied with the revised version.